# Improved DDIM Sampling with Moment Matching Gaussian Mixtures

**Prasad Gabbur**  *pgabbur@gmail.com*
*Independent Researcher**

**Reviewed on OpenReview:** *https://openreview.net/forum?id=CdSPjfmrQN*

## Abstract

We propose using a Gaussian Mixture Model (GMM) as reverse transition operator (kernel) within the Denoising Diffusion Implicit Models (DDIM) framework, which is one of the most widely used approaches for accelerated sampling from pre-trained Denoising Diffusion Probabilistic Models (DDPM). Specifically we match the first and second order central moments of the DDPM forward marginals by constraining the parameters of the GMM. We see that moment matching is sufficient to obtain samples with equal or better quality than the original DDIM with Gaussian kernels. We provide experimental results with unconditional models trained on CelebAHQ and FFHQ, class-conditional models trained on ImageNet, and text-to-image generation using Stable Diffusion v2.1 on COYO700M datasets respectively. Our results suggest that using the GMM kernel leads to significant improvements in the quality of the generated samples when the number of sampling steps is small, as measured by FID and IS metrics. For example on ImageNet 256x256, using 10 sampling steps, we achieve a FID of 6.94 and IS of 207.85 with a GMM kernel compared to 10.15 and 196.73 respectively with a Gaussian kernel. Further, we derive novel SDE samplers for rectified flow matching models and experiment with the proposed approach. We see improvements using both 1-rectified flow and 2-rectified flow models. Code: `https://github.com/pgabbur/ddim-gmm`

## 1 Introduction

Diffusion models (Song & Ermon, 2019; Ho et al., 2020; Song et al., 2021), also known as Score based generative models (Sohl-Dickstein et al., 2015; Song et al., 2020), have demonstrated great success in modeling data distributions in various domains including images (Dhariwal & Nichol, 2021; Nichol & Dhariwal, 2021; Saharia et al., 2022; Rombach et al., 2022), videos (Ho et al., 2022; Blattmann et al., 2023), speech (Kong et al., 2021) and 3D (Poole et al., 2022; Watson et al., 2022). This is due to their flexibility in modeling complex multimodal distributions and ease of training relative to other competitive approaches such as VAEs (Kingma & Welling, 2014; Rezende et al., 2014), GANs (Goodfellow et al., 2014; Salimans et al., 2016; Karras et al., 2018; Brock et al., 2019), autoregressive models (van den Oord et al., 2016b;a) and normalizing flows (Rezende & Mohamed, 2015; Dinh et al., 2017), which do not exhibit both of the advantages simultaneously. In spite of their success, the main bottleneck to their adoption is the slow sampling speed, usually requiring hundreds to thousands of denoising steps to generate a sample. A number of accelerated sampling approaches have emerged, notably Denoising Diffusion Implicit Models (Song et al., 2021), pseudo numerical methods (Liu et al., 2022), distillation (Salimans & Ho, 2022; Luhman & Luhman, 2021; Meng et al., 2023; Yin et al., 2024; Sauer et al., 2024), consistency models (Song et al., 2023; Kim et al., 2023) and various approximations to solving the reverse SDE (Song et al., 2020; Lu et al., 2022; 2023; Zhang & Chen, 2023).

Denoising Diffusion Implicit Models (DDIM) (Song et al., 2021) accelerate sampling from Denoising Diffusion Probabilistic Models (DDPM) (Ho et al., 2020) by hypothesizing a family of non-Markovian forward processes, whose reverse process (Markovian) estimators can be trained with the same surrogate objective

---
*Work done while at Apple.

as DDPMs, assuming the same parameterization for reverse estimators. It is based on the observation that a simplified DDPM training objective $\mathcal{L}_{simple,w}$ (Eq. 4) depends on the forward process only through its marginals at discrete steps $t$. In other words, one can sample with a pretrained DDPM denoiser by designing a different forward/backward process than the original DDPM given that the forward marginals are the same. It has been shown (Xiao et al., 2022; Guo et al., 2023) that the true denoising conditional distributions of DDPMs are multimodal, especially when the denoising step sizes are large. Although the unimodal Gaussian kernel in DDIM yields a multimodal denoising conditional distribution, there is potential to improve its expressiveness with a multimodal kernel. It is not straightforward to use a multimodal kernel while satisfying the marginal constraints. The work of Watson et al. (2021) shows that it is not necessary to satisfy the marginal constraints to achieve accelerated sampling. Taking inspiration from these works, we propose using Gaussian mixtures as the transition kernels of the reverse process in the DDIM framework. This results in a non-Markovian inference process with Gaussian mixtures as marginals. Further, we constrain the mixture parameters so that the first and second order central moments of the forward marginals match exactly those of the DDPM forward marginals. For brevity, we refer to *central moments* as simply *moments* in the rest of the paper.

The work on stochastic interpolants (Albergo et al., 2023) unifies diffusion and rectified flow matching (Liu et al., 2023; Lipman et al., 2023) into a common framework. It has been shown (Gao et al., 2024) that the deterministic Euler and DDIM samplers for rectified flow models are the one and the same. We build upon this equivalence and derive novel SDE samplers for rectified flow matching, which allows us to use the proposed approach within that framework. In summary, our main contributions are as follows:

1. We propose using Gaussian Mixture Models (GMM) as reverse transition operators (kernel) within the DDIM framework, which results in a non-Markovian inference process with Gaussian mixtures as marginals.

2. We derive constraints to match the first and second order moments of the resulting forward GMM marginals to those of the DDPM forward marginals. Based on these constraints, we provide three different schemes to compute GMM parameters efficiently.

3. We demonstrate experimentally that the proposed method results in further accelerating sampling from pretrained DDPM models relative to DDIM, especially with fewer sampling steps.

4. We derive novel SDE solvers for rectified flow matching models and extend them with the proposed approach demonstrating improvements.

We begin by providing a brief overview of Diffusion and the DDIM framework in Section 2. Our approach for extending DDIMs with Gaussian mixture transition kernels is provided in Section 3. We discuss the proposed approach in the context of rectified flow matching in Sec. 4. In Section 5, we conduct experiments on CelebAHQ (Liu et al., 2015), FFHQ (Karras et al., 2019), ImageNet (Deng et al., 2009), and text-to-image generation with Stable Diffusion (Rombach et al., 2022), and provide quantitative results. We also report quantitative results of sampling from CIFAR-10 and ImageNet64 using 1-rectified flow and 2-rectified flow models respectively. Prior work on accelerated sampling is discussed in Section 6. Finally we conclude in Section 7.

## 2 Background

### 2.1 Denoising Diffusion Probabilistic Models

Denoising Diffusion Probabilistic Models (Ho et al., 2020) learn a model for the data distribution $q(\boldsymbol{x}_0)$ by designing a forward and backward diffusion process. In the forward process, noise is added to the data samples following a predetermined schedule, thereby transforming a structured distribution $q(\boldsymbol{x}_0)$ at step $t = 0$ to Gaussian noise, $q(\boldsymbol{x}_T) = \mathcal{N}(\boldsymbol{0}, \boldsymbol{I})$, at step $T$. This is achieved by setting up a Markov chain at every

step $t$ with the transition kernel defined by

$$q(\boldsymbol{x}_t|\boldsymbol{x}_{t-1}) = \mathcal{N}\left(\sqrt{\frac{\alpha_t}{\alpha_{t-1}}}\boldsymbol{x}_{t-1}, \left(1 - \frac{\alpha_t}{\alpha_{t-1}}\right)\boldsymbol{I}\right),\tag{1}$$

where $\alpha_t$ are chosen such that the marginal $q(\boldsymbol{x}_T)$ converges to $\mathcal{N}(\boldsymbol{0}, \boldsymbol{I})$ with large enough $T$. It is straightforward to obtain the marginal of the latent $\boldsymbol{x}_t$ at any step $t$ conditioned on data sample $\boldsymbol{x}_0$ as

$$q(\boldsymbol{x}_t|\boldsymbol{x}_0) = \mathcal{N}\left(\sqrt{\alpha_t}\boldsymbol{x}_0, (1 - \alpha_t)\boldsymbol{I}\right).\tag{2}$$

In the backward process, a parameterized Markovian denoiser $p_\theta(\boldsymbol{x}_{t-1}|\boldsymbol{x}_t)$, initialized with Gaussian noise, $p(\boldsymbol{x}_T) = \mathcal{N}(\boldsymbol{0}, \boldsymbol{I})$, learns to estimate the distribution of $\boldsymbol{x}_{t-1}$ given $\boldsymbol{x}_t$ to maximize the evidence lower bound (ELBO) or minimize $\mathcal{L}_{ELBO}$:

$$\begin{aligned}\mathcal{L}_{ELBO} = \mathbb{E}_q\Big[&D_{KL}(q(\boldsymbol{x}_T|\boldsymbol{x}_0)||p_\theta(\boldsymbol{x}_T))\\ &+ \sum_{t=2}^{T} D_{KL}(q(\boldsymbol{x}_{t-1}|\boldsymbol{x}_t, \boldsymbol{x}_0)||p_\theta(\boldsymbol{x}_{t-1}|\boldsymbol{x}_t))\\ &- \log p_\theta(\boldsymbol{x}_0|\boldsymbol{x}_1)\Big].\end{aligned}\tag{3}$$

The above is equivalent to training $p_\theta(\boldsymbol{x}_{t-1}|\boldsymbol{x}_t)$ to match the posterior $q(\boldsymbol{x}_{t-1}|\boldsymbol{x}_t, \boldsymbol{x}_0)$, which turns out to be a Gaussian (Luo, 2022). Assuming a Gaussian form for $p_\theta(\boldsymbol{x}_{t-1}|\boldsymbol{x}_t) = \mathcal{N}(\boldsymbol{x}_{t-1}|\boldsymbol{\mu}_\theta(\boldsymbol{x}_t, t), \boldsymbol{\Sigma}_\theta(\boldsymbol{x}_t, t))$, leads to a simplified loss function for training DDPMs, which is optimized over $\boldsymbol{\mu}_\theta(\boldsymbol{x}_t, t)$ and $\boldsymbol{\Sigma}_\theta(\boldsymbol{x}_t, t)$. It has been found that a reparameterized mean estimator $\boldsymbol{\mu}_\theta(\boldsymbol{x}_t, t)$ as a function of added noise estimator $\boldsymbol{\epsilon}_\theta(\boldsymbol{x}_t, t)$ has benefits of better sample quality. The resulting simplified loss function $\mathcal{L}_{simple,w}$ (Ho et al., 2020) is a weighted version of ELBO with weights $w_t$:

$$\mathcal{L}_{simple,w} = \mathbb{E}_{t\sim U[1,T]q(\boldsymbol{x}_0)q(\boldsymbol{x}_t|\boldsymbol{x}_0)\boldsymbol{\epsilon}\sim\mathcal{N}(\boldsymbol{0},\boldsymbol{I})}\left[w_t||\boldsymbol{\epsilon} - \boldsymbol{\epsilon}_\theta(\boldsymbol{x}_t, t)||^2\right].\tag{4}$$

$\boldsymbol{\epsilon}_\theta(\boldsymbol{x}_t, t)$ is modeled as a U-Net (Ho et al., 2020) or a transformer (Peebles & Xie, 2022). The covariance estimator $\boldsymbol{\Sigma}_\theta(\boldsymbol{x}_t, t)$ is either learned (Nichol & Dhariwal, 2021; Dhariwal & Nichol, 2021) or fixed (Ho et al., 2020).

## 2.2  DDIM

Following the same notation as Song et al. (2021), we refer to the forward process as *inference* and the reverse process as *generative*. The key assumption is that the more general non-Markovian inference processes $q_\sigma(\boldsymbol{x}_{0:T})$ have the same marginal distribution $q_\sigma(\boldsymbol{x}_t|\boldsymbol{x}_0)$ at every $t$ as the DDPM, but not necessarily the same joint distribution over all the latents $q_\sigma(\boldsymbol{x}_{1:T}|\boldsymbol{x}_0)$. The form of a Markovian family of generative process in DDIMs (Song et al., 2021) is given by

$$\begin{aligned}q_\sigma(\boldsymbol{x}_{1:T}|\boldsymbol{x}_0) &:= q_\sigma(\boldsymbol{x}_T|\boldsymbol{x}_0)\prod_{t=2}^{t=T} q_\sigma(\boldsymbol{x}_{t-1}|\boldsymbol{x}_t, \boldsymbol{x}_0),\\ q_\sigma(\boldsymbol{x}_T|\boldsymbol{x}_0) &:= \mathcal{N}\left(\sqrt{\alpha_T}\boldsymbol{x}_0, (1 - \alpha_T)\boldsymbol{I}\right),\end{aligned}\tag{5}$$

where $\sigma \in R_{\geq 0}^T$ parameterizes the variances of the reverse transition kernels,

$$\begin{aligned}q_\sigma(\boldsymbol{x}_{t-1}|\boldsymbol{x}_t, \boldsymbol{x}_0) &= \mathcal{N}\left(\boldsymbol{\mu}_t(\boldsymbol{x}_t, \boldsymbol{x}_0), \sigma_t^2\boldsymbol{I}\right), \forall t > 1,\\ \boldsymbol{\mu}_t(\boldsymbol{x}_t, \boldsymbol{x}_0) &= \sqrt{\alpha_{t-1}}\boldsymbol{x}_0 + \sqrt{1 - \alpha_{t-1} - \sigma_t^2}.\frac{\boldsymbol{x}_t - \sqrt{\alpha_t}\boldsymbol{x}_0}{\sqrt{1 - \alpha_t}}.\end{aligned}\tag{6}$$

The transition kernel above ensures that the resulting marginals $q_\sigma(\boldsymbol{x}_t|\boldsymbol{x}_0)$ are identical to the DDPM marginals in Eq. 2. The ODE perspective of DDIM and other related works on accelerated sampling from diffusion models are discussed in Section 6.

## 3 Approach

We propose using a Gaussian Mixture Model (GMM) within the reverse transition kernels of the DDIM generative process. Specifically, the form of transition kernels in Eq. 6 is given by

$$q_{\sigma,\mathcal{M}}(\boldsymbol{x}_{t-1}|\boldsymbol{x}_t, \boldsymbol{x}_0) = \sum_{k=1}^{K} \pi_t^k \mathcal{N}(\boldsymbol{\mu}_t^k, \boldsymbol{\Sigma}_t^k), \forall t > 1,$$

$$\boldsymbol{\mu}_t^k = \boldsymbol{\mu}_t(\boldsymbol{x}_t, \boldsymbol{x}_0) + \boldsymbol{\delta}_t^k,$$

$$\boldsymbol{\Sigma}_t^k = \sigma_t^2 \boldsymbol{I} - \boldsymbol{\Delta}_t^k, \tag{7}$$

where $\mathcal{M}_t = \left(\pi_t^k, \boldsymbol{\delta}_t^k, \boldsymbol{\Delta}_t^k\right), k = 1 \dots K$ denote the additional GMM parameters, specifically the mixture component priors, mean and covariance offsets relative to the single Gaussian counterparts of Eq. 6, respectively. Further, we constrain the above kernel so that the first and second order moments of the individual latent variables $q_{\sigma,\mathcal{M}}(\boldsymbol{x}_t|\boldsymbol{x}_0)$ are the same as that of an equivalently parameterized DDPM (Eq. 2). This allows us to use DDIM sampling on a model trained with the same surrogate objective as the DDPM in Eq. 4 given that the GMM parameters $\mathcal{M}_t$ satisfy:

$$\sum_{k=1}^{K} \pi_t^k = 1, \ \ \sum_{k=1}^{K} \pi_t^k \boldsymbol{\delta}_t^k = 0,$$

$$\boldsymbol{\Delta}_t^k = \boldsymbol{\delta}_t^k(\boldsymbol{\delta}_t^k)^T \ \ \text{OR} \ \ \boldsymbol{\Delta}_t^k = \frac{1}{K\pi_t^k} \sum_{l=1}^{K} \pi_t^l \boldsymbol{\delta}_t^l(\boldsymbol{\delta}_t^l)^T, \tag{8}$$

where either one of the two constraints on the covariance matrix offset $\boldsymbol{\Delta}_t^k$ is sufficient to yield the correct moment matching. Please see Appendix A.1 for proof. We also provide an upper bound for the ELBO loss using the proposed inference process as an augmented version of the $\mathcal{L}_{simple,w}$ loss in Appendix A.4. The proposed kernel yields a more expressive multimodal denoising conditional distribution $q_{\sigma,\mathcal{M}}(\boldsymbol{x}_{t-1}|\boldsymbol{x}_t)$ compared to DDIM as shown in Section 3.2.

### 3.1 GMM Parameters

Sampling with the DDIM kernel of Eq. 6 requires choosing an appropriate value for the variance $\sigma_t^2$, which determines the *stochasticity* (Song et al., 2021) of the DDIM inference and sampling processes. It is specified as a proportion $\eta$ of the DDPM reverse transition kernel's variance at the corresponding step $t$. The proposed approach requires choosing the additional GMM parameters $\mathcal{M}_t$ at every step $t$ during sampling. In what follows, we describe three different ways to choose these parameters efficiently, without any training, to satisfy the constraints in Eq. 8 while keeping additional computational requirements relatively low.

First we choose the mixture priors $\pi_t^k$ to be uniform or with a suitable random initialization so that they are non-negative and sum to one. We experiment with choosing the mean offsets $\boldsymbol{\delta}_t^k$ either randomly (DDIM-GMM-RAND) or followed by orthogonalization (DDIM-GMM-ORTHO) to allow for better exploration of the latent space $(\boldsymbol{x}_t, t > 0)$ as described below.

#### 3.1.1 Method 1: DDIM-GMM-RAND

At every step $t$ we sample random vectors $\boldsymbol{o}_t^k, k = 1 \dots K$ from an isotropic multivariate Gaussian with dimensionality equal to that of the latent variables $\boldsymbol{x}_t \in R^D$. These vectors are mean centered and scaled to yield the offsets $\boldsymbol{\delta}_t^k$.

$$\boldsymbol{O}_t \sim \mathcal{N}(\boldsymbol{0}, \boldsymbol{I}), \boldsymbol{O}_t \in R^{D \times K}, K < D,$$

$$\bar{\boldsymbol{o}}_t = \sum_{k=1}^{K} \pi_t^k \boldsymbol{O}_t[k], \ \ \boldsymbol{C}_t[k] = \boldsymbol{O}_t[k] - \bar{\boldsymbol{o}}_t,$$

$$\boldsymbol{\delta}_t^k = \frac{s}{||\boldsymbol{C}_t[k]||_2} \boldsymbol{C}_t[k], \tag{9}$$

where $\boldsymbol{O}_t[k]$ denotes the $k$th column of the matrix $\boldsymbol{O}_t$, $||\boldsymbol{C}_t[k]||_2$ denotes the magnitude of $\boldsymbol{C}_t[k]$, and $s$ is a scale factor that controls the magnitude of the offsets.

### 3.1.2 Method 2: DDIM-GMM-ORTHO

In order to allow for better exploration of the latent space of $\boldsymbol{x}_t$, the set of offsets above is orthonormalized using an SVD on the matrix $\boldsymbol{O}_t$ with $\boldsymbol{o}_t^k$ as columns and choosing the first $K$ components of the output $\boldsymbol{U}_t$ factor, i.e. the first $K$ eigenvectors of $\boldsymbol{O}_t\boldsymbol{O}_t^T$. Specifically

$$\boldsymbol{U}_t\boldsymbol{\Sigma}_t\boldsymbol{V}_t^T = SVD(\boldsymbol{O}_t)$$
$$\bar{\boldsymbol{u}}_t = \sum_{k=1}^{K}\pi_t^k\boldsymbol{U}_t[k], \;\; \boldsymbol{C}_t[1:K] = \boldsymbol{U}_t[1:K] - \bar{\boldsymbol{u}}_t,$$
$$\boldsymbol{\delta}_t^k = s\boldsymbol{C}_t[k], \tag{10}$$

where $\boldsymbol{U}_t[1:K]$ are the first $K$ columns of $\boldsymbol{U}_t$. The mean centering above ensures that the offsets $\boldsymbol{\delta}_t^k$ satisfy the constraint in Eq. 8. The covariance parameters are chosen as

$$\boldsymbol{\Delta}_t^k = \frac{1}{K\pi_t^k}\sum_{l=1}^{K}\pi_t^l\boldsymbol{\delta}_t^l(\boldsymbol{\delta}_t^l)^T \tag{11}$$

to satisfy the covariance constraint of Eq. 8. To sample from the reverse kernel of Eq. 7, we approximate the covariance matrix $\sigma_t^2\boldsymbol{I} - \boldsymbol{\Delta}_t^k$ to be diagonal. A straightforward approximation is to choose only the diagonal elements of $\boldsymbol{\Delta}_t^k$ and subtract from $\sigma_t^2$.

### 3.1.3 Method 3: DDIM-GMM-ORTHO-VUB

Given the random choice of offsets $\boldsymbol{\delta}_t^k$, we also experiment with an upper bound diagonal approximation of $\boldsymbol{\Delta}_t^k$ by eigen decomposition similar to PCA (Jolliffe, 1986), These variance upper bounds (VUB) determine the maximum allowable variances for the dimensions in $\boldsymbol{\Delta}_t^k$ keeping the total variance the same. If $\lambda_i, i = 1\ldots K$ are the eigenvalues of $\boldsymbol{\Delta}_t^k$, then

$$\frac{s^2}{K\pi_t^k}\left(\pi_t^1 - \sum_{l=1}^{K}(\pi_t^l)^2\right) \le \lambda_1 \le \frac{s^2}{K\pi_t^k}\pi_t^1,$$
$$\frac{s^2}{K\pi_t^k}\pi_t^{i-1} \le \lambda_i \le \frac{s^2}{K\pi_t^k}\pi_t^i, \quad\quad i = 2\ldots K. \tag{12}$$

Please see Appendix A.2 for proof. It turns out the upper bounds above are independent of the $\boldsymbol{\delta}_t^k$'s. We use the upper bounds in Eq. 34 to compute the diagonal approximation of $\sigma_t^2\boldsymbol{I} - \boldsymbol{\Delta}_t^k$ by offsetting the first $K$ elements of $\sigma_t^2\boldsymbol{I}$ with the eigenvalue upper bounds. The scale $s$ can be chosen such that the upper bounds are always smaller than $\sigma_t^2$ to ensure positive variances across all dimensions. Note that the above diagonalization and variance offsetting has to be done only once before sampling, which introduces additional computation for initialization but not during sampling. We also experiment with sharing GMM parameters across sampling steps $t$ to save time by avoiding the expensive SVD operation for each step and doing it only once. We denote this approach as DDIM-GMM-ORTHO-VUB$^*$. Please see Appendix A.10 for further discussion on additional computational overhead.

## 3.2 DDIM-GMM as a Multimodal Denoiser

Recent works (Guo et al., 2023; Xiao et al., 2022) have shown that the target conditional distribution $q(\boldsymbol{x}_{t-1}|\boldsymbol{x}_t)$, to be estimated by the denoiser $p_\theta(\boldsymbol{x}_{t-1}|\boldsymbol{x}_t)$, is multimodal in real-world datasets. Here we show that the proposed DDIM-GMM sampling scheme addresses the unimodal assumption of the single Gaussian denoisers in pre-trained diffusion models better than DDIM. We start by showing that the proposed DDIM-GMM kernel yields a multimodal conditional distribution $q_{\sigma,\mathcal{M}}(\boldsymbol{x}_{t-1}|\boldsymbol{x}_t)$. Let the true data distribution

$q(\boldsymbol{x}_0)$ be a Dirac distribution given by

$$q(\boldsymbol{x}_0) = \sum_i w_i \delta(\boldsymbol{x}_0 - \boldsymbol{x}_0^i), \tag{13}$$

where $\boldsymbol{x}_0^i$ are the observed data points. The DDIM-GMM denoiser's conditional distribution $q_{\sigma,\mathcal{M}}(\boldsymbol{x}_{t-1}|\boldsymbol{x}_t)$ can be obtained using Bayes' rule:

$$
\begin{aligned}
q_{\sigma,\mathcal{M}}(\boldsymbol{x}_{t-1}|\boldsymbol{x}_t) &= \int_{\boldsymbol{x}_0} q_{\sigma,\mathcal{M}}(\boldsymbol{x}_{t-1}|\boldsymbol{x}_t, \boldsymbol{x}_0) q_{\sigma,\mathcal{M}}(\boldsymbol{x}_0|\boldsymbol{x}_t)\,\mathrm{d}\boldsymbol{x}_0 \\
&\propto \int_{\boldsymbol{x}_0} q_{\sigma,\mathcal{M}}(\boldsymbol{x}_{t-1}|\boldsymbol{x}_t, \boldsymbol{x}_0) q_{\sigma,\mathcal{M}}(\boldsymbol{x}_t|\boldsymbol{x}_0) q(\boldsymbol{x}_0)\,\mathrm{d}\boldsymbol{x}_0 \\
&= \sum_i w_i q_{\sigma,\mathcal{M}}(\boldsymbol{x}_{t-1}|\boldsymbol{x}_t, \boldsymbol{x}_0^i) q_{\sigma,\mathcal{M}}(\boldsymbol{x}_t|\boldsymbol{x}_0^i),
\end{aligned}
\tag{14}
$$

which is a mixture of Gaussians. This follows from the fact that $q_{\sigma,\mathcal{M}}(\boldsymbol{x}_{t-1}|\boldsymbol{x}_t, \boldsymbol{x}_0^i)$ is a mixture of Gaussians given by Eq. 7 and $q_{\sigma,\mathcal{M}}(\boldsymbol{x}_t|\boldsymbol{x}_0^i)$ is a scalar constant given $\boldsymbol{x}_t$. A similar argument holds even when the data distribution $q(\boldsymbol{x}_0)$ is a mixture of Gaussians. The resulting denoiser is a mixture of Gaussians, whose form can be obtained by noting that $q_{\sigma,\mathcal{M}}(\boldsymbol{x}_t|\boldsymbol{x}_0^i)$ is a GMM and accordingly completing squares within the integrand above (Bishop, 2006). A similar argument as above also enables DDIM sampler to model a multimodal denoising distribution $q_\sigma(\boldsymbol{x}_{t-1}|\boldsymbol{x}_t)$. However the multimodality of the kernel $q_{\sigma,\mathcal{M}}(\boldsymbol{x}_{t-1}|\boldsymbol{x}_t, \boldsymbol{x}_0^i)$ in Eq. 14 enables DDIM-GMM to express more complex denoising distributions than DDIM. This has the potential to better match the unknown distribution $q(\boldsymbol{x}_{t-1}|\boldsymbol{x}_t)$, especially when the number of sampling steps is small (Guo et al., 2023; Xiao et al., 2022), without any training or fine-tuning with specialized loss functions.

## 4 Rectified Flow Implicit Models (RFIM)

Flow matching (Lipman et al., 2023; Albergo et al., 2023; Albergo & Vanden-Eijnden, 2023) methods learn a mapping between a source distribution $(\boldsymbol{x}_0 \sim q_0)$ at $\tau = 0$ and a target distribution $(\boldsymbol{x}_1 \sim q_1)$ at $\tau = 1$ by formulating a stochastic process $(\boldsymbol{x}_\tau \sim p_\tau, \tau \in [0, 1])$ as a bridge between the two distributions. Specifically, rectified flow matching (Liu et al., 2023) defines an optimal transport (OT) path between $\boldsymbol{x}_0$ and $\boldsymbol{x}_1$ as

$$\boldsymbol{x}_\tau = (1 - \tau)\boldsymbol{x}_0 + \tau\boldsymbol{x}_1, \tau \in [0, 1]. \tag{15}$$

In order to transfer samples from source to target distribution or vice versa, a velocity function $\boldsymbol{v}_\theta(\boldsymbol{x}_\tau, \tau)$, conditioned on $\boldsymbol{x}_\tau$ and parameterized by $\theta$, is learnt to match the instantaneous velocity $\frac{d\boldsymbol{x}_\tau}{d\tau} = (\boldsymbol{x}_1 - \boldsymbol{x}_0)$ in expectation by sampling $(\boldsymbol{x}_0, \boldsymbol{x}_1)$ from a pre-defined coupling of $q_0$ and $q_1$, usually the independent coupling $q_0(\boldsymbol{x}_0) * q_1(\boldsymbol{x}_1)$. This is achieved by minimizing an unconstrained least squared error loss within the conditional flow matching framework (Liu et al., 2023):

$$\mathcal{L}_{CFM}(\theta) = \mathbb{E}_{\boldsymbol{x}_0 \sim q_0, \boldsymbol{x}_1 \sim q_1, \tau \sim U(0,1)} \left[ ||(\boldsymbol{x}_1 - \boldsymbol{x}_0) - \boldsymbol{v}_\theta(\boldsymbol{x}_\tau, \tau)||^2 \right], \tag{16}$$

where $\boldsymbol{x}_\tau$ is given by Eq. 15. Starting from an initial sample $\boldsymbol{x}_0$ from the source distribution, solving the following first order ODE using the learned velocity $\boldsymbol{v}_\theta(\boldsymbol{x}_\tau, \tau)$ yields samples from the target distribution at $\tau = 1$:

$$d\boldsymbol{x}_\tau = \boldsymbol{v}_\theta(\boldsymbol{x}_\tau, \tau)d\tau. \tag{17}$$

Usually the ODE is run for a finite number of steps $T$ by discretizing the continuous interval $\tau \in [0, 1]$, for e.g. $\tau = \frac{i}{T}, i = 0 \dots T$. The Euler update in Eq. 17 has been shown to be equivalent to the deterministic DDIM step ($\eta = 0$) in the context of Diffusion models (Gao et al., 2024). Motivated by this equivalence, we propose a family of implicit models for rectified flow matching similar to DDIM for diffusion. The key observation is that the simplified diffusion loss of Eq. 4 and the conditional flow matching loss of Eq. 16 have

been shown to be equivalent by a simple reparameterization of the model and an appropriate choice of the loss weights $w_t$ (Lee et al., 2024; Kim et al., 2025). This leads to a family of parameterized implicit models for sampling, similar to DDIM, whose marginals at every intermediate time step $\tau$ match the corresponding marginals from which the training data was sampled.

Assume that $q_0$ is the true data distribution and $q_1$ is the standard Gaussian $\mathcal{N}(\mathbf{0}, \boldsymbol{I})$. It is straightforward to verify that the marginal of the interpolant $\boldsymbol{x}_\tau$ conditioned on $\boldsymbol{x}_0$ is given by

$$q(\boldsymbol{x}_\tau|\boldsymbol{x}_0) = \mathcal{N}((1-\tau)\boldsymbol{x}_0, \tau^2 \boldsymbol{I}). \tag{18}$$

We define a parameterized Markov chain to sample the interpolants $\boldsymbol{x}_{\tau_i}$ for discrete $\tau_i = \frac{i}{T}, i \in \{0 \dots T\}$ as

$$q_\sigma(\boldsymbol{x}_{\tau_{1:T}}|\boldsymbol{x}_0) := q_\sigma(\boldsymbol{x}_{\tau_T}|\boldsymbol{x}_0) \prod_{i=2}^{i=T} q_\sigma(\boldsymbol{x}_{\tau_{i-1}}|\boldsymbol{x}_{\tau_i}, \boldsymbol{x}_0),$$

$$q_\sigma(\boldsymbol{x}_{\tau_T}|\boldsymbol{x}_0) := \mathcal{N}(\mathbf{0}, \boldsymbol{I}), \tag{19}$$

where $\sigma \in R_{\geq 0}^T$ parameterizes the variances of the Markov chain transition kernels. The transition kernel from any $\tau$ to $s$, $0 \leq s < \tau \leq 1$, is given by

$$q_\sigma(\boldsymbol{x}_s|\boldsymbol{x}_\tau, \boldsymbol{x}_0) = \mathcal{N}\left(\boldsymbol{\mu}_\tau(\boldsymbol{x}_\tau, \boldsymbol{x}_0), \sigma_\tau^2 \boldsymbol{I}\right), \forall 0 \leq s < \tau \leq 1,$$

$$\boldsymbol{\mu}_{\tau_i}(\boldsymbol{x}_\tau, \boldsymbol{x}_0) = \frac{\sqrt{s^2 - \sigma_\tau^2}}{\tau}\boldsymbol{x}_{\tau_i} + \left((1-s) + \frac{(1-\tau)}{\tau}\sqrt{s^2 - \sigma_\tau^2}\right)\boldsymbol{x}_0. \tag{20}$$

Using the above transition kernel, it can be verified that the marginal $q_\sigma(\boldsymbol{x}_s|\boldsymbol{x}_0)$ has the same form as the interpolant marginal of Eq. 18 at $\tau = s$. Therefore samples from the proposed Markov chain come from the same marginal distributions as the interpolant marginals for all $\tau$ conditioned on $\boldsymbol{x}_0$. In practice, we do not have access to $\boldsymbol{x}_0$ but use an estimate from a trained model. Specifically, we use the equivalence between diffusion and rectified flow matching loss (Lee et al., 2024; Kim et al., 2025) and adapt an $\boldsymbol{x}_0(\boldsymbol{x}_\tau, \tau)$ prediction model to yield the velocity $\boldsymbol{v}_\tau(\boldsymbol{x}_\tau, \tau)$. Specifically, following the Rectified Flow++ work (Lee et al., 2024), the velocity $\boldsymbol{v}_\tau(\boldsymbol{x}_\tau)$ can be expressed as

$$\boldsymbol{v}_\tau(\boldsymbol{x}_\tau) = \frac{d\boldsymbol{x}_\tau}{d\tau} := \frac{\boldsymbol{x}_\tau - \mathbb{E}\left[\boldsymbol{x}_0|\boldsymbol{x}_\tau\right]}{\tau}, \tag{21}$$

where a parametric model for predicting $\mathbb{E}\left[\boldsymbol{x}_0|\boldsymbol{x}_\tau\right]$ is trained with the $\mathcal{L}_{CFM}$ loss of Eq. 16 with $\boldsymbol{v}_\theta(\boldsymbol{x}_\tau, \tau)$ being obtained from the reparameterization of Eq. 21.

Note that in order for Eq. 20 to be a valid kernel, $0 \leq \sigma_\tau^2 \leq s^2$, which leads to a couple of interesting special cases at the extremities. If $\sigma_\tau = 0$, the resulting sampler reduces to a discrete version of the Euler ODE sampler of Eq. 17. At the other extreme ($\sigma_\tau = s$), the sampler becomes non-Markovian yielding independent samples from the interpolant marginal distribution conditioned on $\boldsymbol{x}_0$, i.e., $q_\sigma(\boldsymbol{x}_s|\boldsymbol{x}_\tau, \boldsymbol{x}_0) = q(\boldsymbol{x}_s|\boldsymbol{x}_0)$. We experiment with $\sigma_\tau = \eta s$ for a few different values of $\eta \in [0, 1]$ similar to DDIM (Song et al., 2021) for diffusion models.

## 4.1 Moment Matching Gaussian Mixtures (RFIM-GMM)

We extend our proposed approach to rectified flow models by using Markov transition kernels that use Gaussian mixtures in place of unimodal Gaussians. The resulting samplers are denoted with the RFIM-GMM-* notation similar to DDIM-GMM-*. Specifically the form of the RFIM-GMM kernels is the same as Eq. 7 with $t$ replaced by its discrete counterpart $\tau_i \in (0, 1]$. Similarly the constraints of Eq. 8 satisfy the moment matching condition, i.e., the first and second order central moments of the RFIM-GMM sampler marginals $q_{\sigma,\mathcal{M}}(\boldsymbol{x}_{\tau_i}|\boldsymbol{x}_0)$ match the corresponding moments of the interpolant marginals $q(\boldsymbol{x}_{\tau_i}|\boldsymbol{x}_0)$ for each $\tau_i$. We experiment with three different choices for choosing the mean offsets of the Gaussian mixture kernels described in Sec. 3.1 and refer to them as RFIM-GMM-RAND, RFIM-GMM-ORTHO and RFIM-GMM-ORTHO-VUB respectively.

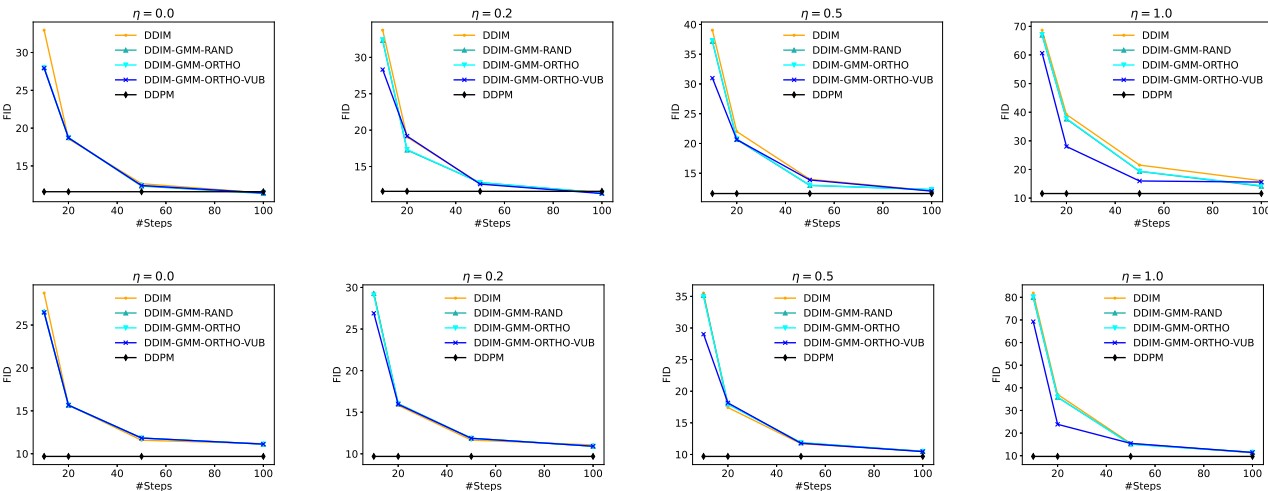

Figure 1: **CelebAHQ** (top) and **FFHQ** (bottom). FID ($\downarrow$). The horizontal line is the DDPM baseline run for 1000 steps.

We show that the RFIM-GMM kernels provide more expressivity relative to their unimodal counterparts (RFIM) in modeling the true denoising distributions $q(\boldsymbol{x}_{\tau_{i-1}}|\boldsymbol{x}_{\tau_i}), i \in \{1 \ldots T\}$ within the rectified flow matching framework. This can be seen by examining the form of the denoising distributions between two successive discretized interpolant time points $(\tau_{i-1}, \tau_i)$ as

$$
\begin{aligned}
q(\boldsymbol{x}_{\tau_{i-1}}|\boldsymbol{x}_{\tau_i}) &= \int_{\boldsymbol{x}_0} q(\boldsymbol{x}_{\tau_{i-1}}|\boldsymbol{x}_{\tau_i}, \boldsymbol{x}_0) q(\boldsymbol{x}_0|\boldsymbol{x}_{\tau_i}) \, \mathrm{d}\boldsymbol{x}_0 \\
&\propto \int_{\boldsymbol{x}_0} \delta(\boldsymbol{x}_{\tau_{i-1}} - \boldsymbol{x}_{\tau_i} - (\boldsymbol{x}_{\tau_i} - \boldsymbol{x}_0)(\tau_{i-1} - \tau_i)) q(\boldsymbol{x}_{\tau_i}|\boldsymbol{x}_0) q(\boldsymbol{x}_0) \, \mathrm{d}\boldsymbol{x}_0 \\
&\propto \int_{\boldsymbol{x}_0} \delta(\boldsymbol{x}_{\tau_{i-1}} - \boldsymbol{x}_{\tau_i} - (\boldsymbol{x}_{\tau_i} - \boldsymbol{x}_0)(\tau_{i-1} - \tau_i)) \exp\left(-\frac{||\boldsymbol{x}_{\tau_i} - (1 - \tau_i)\boldsymbol{x}_0||^2}{\tau_i^2}\right) q(\boldsymbol{x}_0) \, \mathrm{d}\boldsymbol{x}_0, \quad (22)
\end{aligned}
$$

where we have made use of Eq. 18 to arrive at Eq. 22. In general, the true data distribution $q(\boldsymbol{x}_0)$ is multimodal (e.g. Eq. 13) implying a complex multimodal form of the true denoising distribution. Using the same arguments as Sec. 3.2, RFIM-GMM kernels enable more flexibility in modeling $q(\boldsymbol{x}_{\tau_{i-1}}|\boldsymbol{x}_{\tau_i})$ relative to RFIM kernels.

## 5 Experiments

In this section we compare the quality of samples generated using the proposed approach with those generated by the original DDIM sampling. Using diffusion models, we conduct experiments on CelebAHQ (Liu et al., 2015) and FFHQ (Karras et al., 2019), which are high resolution face datasets used as standard benchmarks for evaluating generative models. We also evaluate the effectiveness of the proposed approach on sampling from class-conditional distributions by training on the ImageNet dataset with conditioning on class labels (Rombach et al., 2022). The sample quality is measured using Frechét Inception Distance (FID) (Heusel et al., 2017) and Inception score (IS) (Salimans et al., 2016) for class-conditional generation. We train diffusion models using the unweighted DDPM objective (Ho et al., 2020) in the latent space of a VQVAE (Rombach et al., 2022). More experimental details can be found in the Appendix A.5. For each dataset, we generate as many samples as in the standard validation split of the dataset to compute the FID and IS metrics, i.e., 5000 for CelebAHQ, 10000 for FFHQ and 50000 for ImageNet respectively. Experimental results on text-to-image generation using Stable Diffusion v2.1 on the COYO700M dataset (Byeon et al., 2022) are reported in Sec. 5.3. We also report experimental results with rectified flow matching models in Sec. 5.4. Specifically, we compare RFIM vs. RFIM-GMM samplers on the FID metric computed using

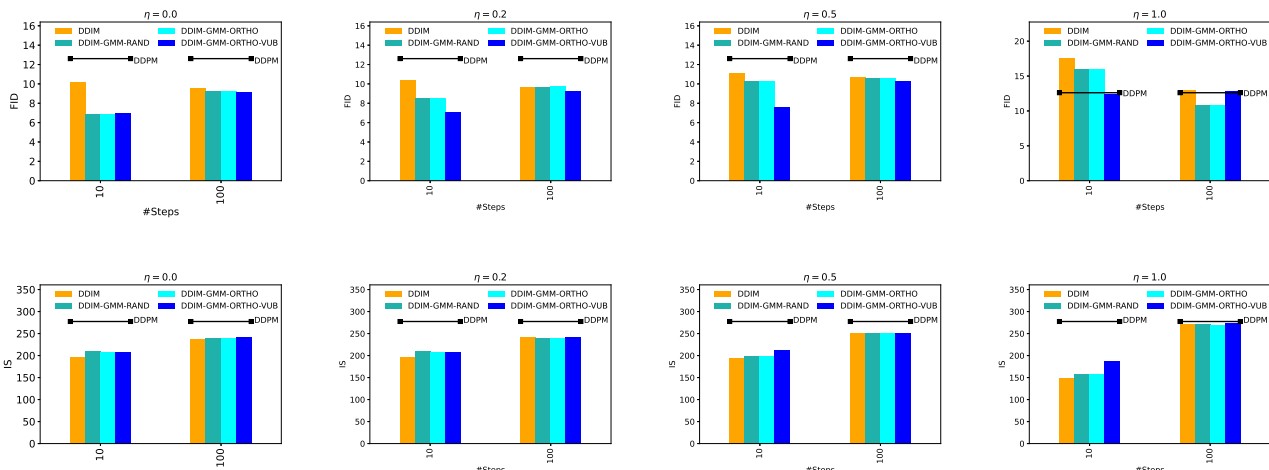

Figure 2: **Class-conditional ImageNet with Classifier-free Guidance**. FID ($\downarrow$) (top) and IS ($\uparrow$) (bottom) on the 50k ImageNet validation set. Classifier-free guidance with a scale of 2.5 is used during inference. The horizontal line is the DDPM baseline run for 1000 steps.

50000 samples generated from 1-rectified flow and 2-rectified flow models on CIFAR10 (Krizhevsky, 2009) and ImageNet64 (Deng et al., 2009) datasets respectively.

## 5.1 Unconditional Models on CelebAHQ and FFHQ

The FID scores of unconditional generation models on CelebAHQ and FFHQ datasets are reported in Fig. 1. We run both DDIM and the proposed variants of DDIM-GMM samplers for different numbers of steps (10, 20, 50, 100) using different values of the stochasticity parameter $\eta$ (Song et al., 2021). For each DDIM-GMM variant, we choose a GMM with 8 mixture components with uniform priors ($\pi_t^k$=0.125) for all steps $t$. We also search for the best value of scaling $s$ among $\{0.01, 0.1, 1.0, 10.0\}$ and report the best result with the chosen value for $s$. For all the experiments here, we set the value of $s$ to be the same for all steps $t$. It is possible to further tune these parameters. For instance one could search for an optimal set of parameters using a suitable objective (Watson et al., 2021; Mathiasen & Hvilshøj, 2021) on the training set. We also run full DDPM sampling for 1000 steps as a baseline since all the models were trained for 1000 steps in the forward process using the $\mathcal{L}_{simple,w}$ DDPM objective (Eq. 4) with uniform weights ($w = \mathbf{1}$). We observe that sampling with a GMM transition kernel (DDIM-GMM-*) shows significant improvements in sample quality over the Gaussian kernel (DDIM) at lower values of sampling steps and higher values of $\eta$ for unconditional generation on both CelebAHQ and FFHQ (see also Tables 8 and 9, Appendix A.14). Among the different choices for computing GMM offset parameters, DDIM-GMM-RAND and DDIM-GMM-ORTHO produce similar quality results. We observe significant improvements with upper bounding variances with the DDIM-GMM-ORTHO-VUB variant. Our hypothesis is that the GMM kernel allows exploring the latent space better than the Gaussian kernel under those settings. Variance upper bounding further encourages this by lumping variances into fewer dimensions of $\boldsymbol{\Delta}_t^k$. This is favorable since sampling time is a significant bottleneck for the use of DDPM in real-time applications.

## 5.2 Class-conditional ImageNet

We train class-conditional models on ImageNet and experiment with guided sampling using classifier-free guidance (Ho & Salimans, 2021). Specifically, we jointly train a class-conditional and unconditional model with parameter sharing (Ho & Salimans, 2021) by setting the unconditional training probability to 0.1. We then sample from this model using a guidance scale of 2.5 for 10 and 100 sampling steps. Note that each sampling step involves two Neural Function Evaluations (NFE) using classifier-free guidance in order to

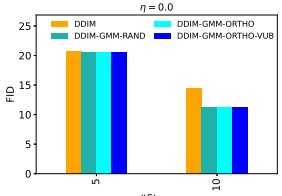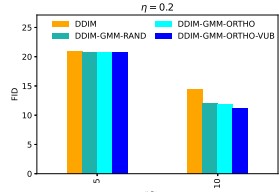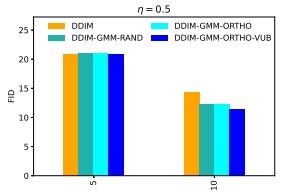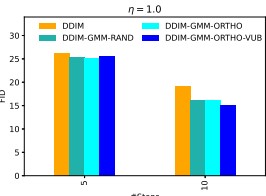

Figure 3: **Text-to-Image Generation**. FID ($\downarrow$) on a 30k subset of COYO-700M using the Stable Diffusion v2.1 model. Classifier-free guidance with a scale of 7.5 is used during inference.

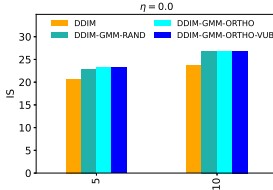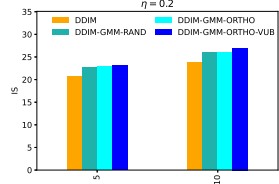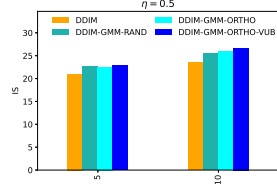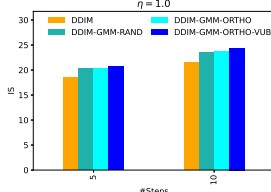

Figure 4: **Text-to-Image Generation**. IS ($\uparrow$) on a 30k subset of COYO-700M using the Stable Diffusion v2.1 model. Classifier-free guidance with a scale of 7.5 is used during inference.

compute conditional and unconditional scores with the same denoising network. The FID and IS results are shown in Fig. 2. Using fewer sampling steps (10), the FID and IS scores of the samples improve significantly when any DDIM-GMM-* sampler is used, relative to DDIM (see Tables 12 and 13, Appendix A.14.3). Notably, the deterministic ($\eta = 0$) DDIM-GMM-ORTHO-VUB* sampler and the DDIM-GMM-ORTHO-VUB sampler at $\eta = 0.5$ yield the best FID (6.72) and IS (211.93) respectively. Similar to previous results, among DDIM-GMM-*, using variance bounding leads to more significant improvements at higher $\eta$ relative to other variants. Using 100 steps, samples from DDIM and DDIM-GMM-* variants have similar metrics in most settings with some exceptions. DDIM-GMM-RAND and DDDIM-GMM-ORTHO yield significantly better FID values than DDIM under the $\eta = 1$ setting. DDIM-GMM-ORTHO-VUB samples have consistently higher IS values under all settings and yield the best FID (9.13) at $\eta = 0$. We posit that the similar performance of DDIM and DDIM-GMM-* samplers using larger number of steps is due to the possibility that the multimodality of the true denoiser conditional distribution ($q(x_{t-1}|x_t)$) is modeled equally well by both the samplers (Guo et al., 2023; Xiao et al., 2022). See Appendix A.6 for additional results using classifier-guidance. Also, Appendix A.13 shows some qualitative results of sampling with the proposed approach compared to DDIM.

## 5.3 Text-to-Image Generation

In this section, we experiment with a pretrained text-to-image diffusion model. Specifically, we use the publicly available Stable Diffusion v2.1 (Rombach et al., 2022) on a subset of 30,000 text and image pairs from the large scale COYO-700M image-text pair dataset (Byeon et al., 2022). Stable Diffusion v2.1 is a text-to-image diffusion model conditioned on text captions. It is trained on a subset of the large-scale LAION-5B image-text pair dataset (Schuhmann et al., 2022). We use DDIM and the variants of DDIM-GMM samplers, each with 5 and 10 sampling steps, to generate images at 256x256 resolution conditioned on captions from the COYO-700M data subset. Fig. 3 and 4 show the FID and IS metrics respectively. The FID metric improves consistently with the DDIM-GMM-* samplers relative to DDIM for all settings of $\eta$ using 10 sampling steps. The relative improvements with DDIM-GMM-ORTHO-VUB over DDIM are more significant with increasing $\eta$ compared to DDIM-GMM-RAND and DDIM-GMM-ORTHO suggesting better exploration of latent space, similar to results with unconditional models. Using 5 sampling steps, the DDIM-GMM-* samplers show most improvements with $\eta = 1$. On the IS metric, all variants of DDIM-GMM

samplers show significant improvements over DDIM under different settings of $\eta$ and sampling steps. Fig. 12 in Appendix A.13 shows some sample generated images.

## 5.4 Sampling from Rectified Flow Matching models

We report quantitative results by sampling from rectified flow matching models using the proposed RFIM and RFIM-GMM kernels. We experiment with both 1-rectified flow and 2-rectified flow matching models and compute the FID (Heusel et al., 2017) metric to quantify sample quality. Specifically, we train a 1-rectified flow model on the CIFAR-10 dataset using the UNet architecture and OT-CFM loss (Tong et al., 2024). For 2-rectified flow, we use the pre-trained ImageNet64 model from Lee et al. (2024) and sample from it. For all our experiments, we choose $\eta \in \{0.0, 0.2, 0.5\}$ as we found that higher values of $\eta$ led to worse sample quality, likely due to the sampling chain becoming progressively non-Markovian and completely independent at $\eta = 1$ (see Sec. 4). Table 1 shows the results of sampling from the 1-rectified flow CIFAR-10 model for up to 50 sampling steps. The RFIM-GMM kernels consistently lead to better sample quality (lower FID) using different values of $\eta$. We also see small improvements with the RFIM-GMM kernels on the pre-trained 2-rectified flow ImageNet64 model when using only 1 or 2 sampling steps. In this case, the relative improvements are the same for all the different RFIM-GMM-* kernels. Interestingly, the additional stochasticity from the offsets of the RFIM-GMM kernels seems to improve the FID relative to the deterministic single-step and Euler ODE (RFIM, $\eta = 0$) samplers.

Table 1: **1-Rectified Flow, CIFAR10**. FID ($\downarrow$)

|  | Steps | 2 | 5 | 10 | 50 |
|---|---|---|---|---|---|
| $\eta$ |  |  |  |  |  |
| 0 | RFIM | 89.08 | 25.20 | 14.13 | 6.37 |
| 0 | RFIM-GMM-RAND | 88.10 | 24.61 | 13.69 | 6.16 |
| 0 | RFIM-GMM-ORTHO | 88.11 | 24.62 | 13.69 | 6.13 |
| 0 | RFIM-GMM-ORTHO-VUB | 88.04 | 24.62 | 13.69 | 6.13 |
| 0.2 | RFIM | 89.66 | 25.87 | 14.41 | 6.52 |
| 0.2 | RFIM-GMM-RAND | 88.65 | 25.17 | 14.20 | 6.27 |
| 0.2 | RFIM-GMM-ORTHO | 88.66 | 25.18 | 14.19 | 6.26 |
| 0.2 | RFIM-GMM-ORTHO-VUB | 88.65 | 25.17 | 14.20 | 6.27 |
| 0.5 | RFIM | 93.15 | 30.02 | 17.90 | 17.67 |
| 0.5 | RFIM-GMM-RAND | 92.01 | 29.03 | 17.59 | 17.32 |
| 0.5 | RFIM-GMM-ORTHO | 92.00 | 29.02 | 17.59 | 17.34 |
| 0.5 | RFIM-GMM-ORTHO-VUB | 92.00 | 29.02 | 17.59 | 17.35 |

Table 2: **2-Rectified Flow, ImageNet64**. FID ($\downarrow$)

|  | Steps | 1 | 2 |
|---|---|---|---|
| $\eta$ |  |  |  |
| 0 | RFIM | 4.42 | 3.97 |
| 0 | RFIM-GMM-* | 4.38 | 3.94 |
| 0.2 | RFIM | 4.42 | 3.99 |
| 0.2 | RFIM-GMM-* | 4.38 | 3.95 |
| 0.5 | RFIM | 4.42 | 4.25 |
| 0.5 | RFIM-GMM-* | 4.38 | 4.20 |

## 6 Related Work

Prior and concurrent work on accelerated sampling for pretrained diffusion models can be broadly categorized into implicit modeling (Song et al., 2021; Zhang et al., 2022; Watson et al., 2021), distillation (Luhman & Luhman, 2021; Salimans & Ho, 2022; Meng et al., 2023; Yin et al., 2024; Sauer et al., 2024), consistency models (Song et al., 2023; Kim et al., 2023) and ODE solver (Song et al., 2020; Jolicoeur-Martineau et al., 2021; Zhang & Chen, 2023; Karras et al., 2022; Lu et al., 2023; Liu et al., 2022) based approaches. Watson et al. (2021) extend DDIMs by introducing a more general family of implicit distributions with learnable parameters trained with backpropagation using perceptual loss. While the proposed approach also introduces learnable parameters, our marginals are Gaussian mixtures and we ensure that the moments are matched exactly with those of the DDPM marginals. Zhang et al. (2022) analyze the workings of DDIM using a limiting case of Dirac distribution in the data space and generalize it to non-isotropic diffusion models. Other approaches propose accelerated sampling by modeling DDPMs with non-Gaussian noise (Nachmani et al., 2021) or learning noise levels of the reverse process separately (San-Roman et al., 2021). Different from these, the proposed approach introduces a different sampling kernel in the reverse process of the DDIM framework (Song et al., 2021). Our work is also complementary to distillation based approaches, which might further benefit from an improved DDIM teacher (Salimans & Ho, 2022; Meng et al., 2023).

By treating sampling as solving reverse direction diffusion ODEs (Song et al., 2020), acceleration is achieved by discretization with linear (Song et al., 2021) or higher order approximations (Jolicoeur-Martineau et al., 2021; Lu et al., 2022; Zhang & Chen, 2023). Being a moment matching version of DDIM, the proposed approach can be thought of as a linear solver. It is observed that higher-order solvers are inherently unstable in the guided sampling regime, especially if the guidance weight is high (Lu et al., 2023). Our empirical results suggest that, even with a high guidance weight, moment matching based DDIM-GMM is beneficial for guided sampling with few sampling steps. Rectified flow matching (Liu et al., 2023; Lipman et al., 2023; Albergo et al., 2023) framework inherently learns to produce straighter sampling paths compared to diffusion models leading to relatively faster sampling using ODE solvers such as the first order Euler or the second order Heun (Lee et al., 2024) solvers. Multiple rectification iterations (Liu et al., 2023; Lee et al., 2024; Kim et al., 2025) further help straighten the paths improving sampling efficiency. Our work proposes novel SDE solvers that includes Euler ODE as a special case.

## 7 Conclusions

We propose improved DDIM sampling by using Gaussian mixture transition kernels whose marginal first and second order moments match the corresponding moments of the DDPM forward marginals. Our experiments suggest that moment matching is sufficient to produce samples of the same or better quality than the original DDIM sampler. This is especially true if the number of sampling steps is small (e.g. 10) using unconditional models trained on CelebAHQ and FFHQ. For guided ImageNet class-conditional models, the GMM kernel based samplers lead to improvements in both FID and IS metrics under almost all settings of $\eta$ and number of sampling steps (10 and 100). This seems to suggest that the GMM kernel allows for a better exploration of the latent space with a small number of sampling steps. We also demonstrate that DDIM-GMM shows improvements over DDIM for few step sampling from text-to-image models. Using the equivalence between diffusion and rectified flow matching, we derive novel SDE samplers for rectified flow models and demonstrate improvements with the proposed approach.

The gap between DDIM and the proposed DDIM-GMM becomes smaller with larger number of steps using training-free GMM parameter selection. An interesting future direction would be to optimize the GMM parameters $\mathcal{M}_t$ to maximize a suitable metric such as KID (Watson et al., 2021) or FID (Mathiasen & Hvilshøj, 2021) on the training dataset.

## Broader Impact

Our work aims to accelerate sampling from pre-trained diffusion and rectified flow matching models by building upon the widely used DDIM framework. On one hand, fast sampling from these models has the potential for positive societal benefits by reducing the computational burden and thereby the carbon footprint resulting from their large-scale deployment. On the other hand, generative models are known to pose risks to society such as aiding misinformation, privacy invasion and phishing. All these have been widely discussed and we wish not to list them in detail here.

## Acknowledgments

We thank Miguel Angel Bautista Martin, Navdeep Jaitly, Ian Fasel and Barry Theobald for their invaluable help in the review and publishing process of this work.

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

# A    Appendix

## A.1    Proof of Constraints on GMM Parameters

Our proof for the constraints in Eq. 8 follows by induction (Song et al., 2021). The marginal of $\boldsymbol{x}_T$ is already equal to the DDPM marginal at step $T$ by definition (Eq. 5). We show below that the marginals of all the random variables $\boldsymbol{x}_t, t < T$ are Gaussian mixtures with their first and second order moments equal to the desired values, given the constraints in Eq. 8. We derive the forms of the marginals for $T-1$ and $T-2$ and the proof follows inductively for all $t < T-2$. Using Bayes' rule, the marginal at $\boldsymbol{x}_{T-1}$ is given by

$$q_{\sigma,\mathcal{M}}(\boldsymbol{x}_{T-1}|\boldsymbol{x}_0) = \int_{\boldsymbol{x}_T} q_{\sigma,\mathcal{M}}(\boldsymbol{x}_{T-1}|\boldsymbol{x}_T,\boldsymbol{x}_0)q_{\sigma,\mathcal{M}}(\boldsymbol{x}_T|\boldsymbol{x}_0)\,\mathrm{d}\boldsymbol{x}_T$$

$$= \int_{\boldsymbol{x}_T} \sum_{k=1}^{K} \pi_T^k \mathcal{N}\left(\sqrt{\alpha_{T-1}}\boldsymbol{x}_0 + \sqrt{1-\alpha_{T-1}-\sigma_T^2}.\frac{\boldsymbol{x}_T - \sqrt{\alpha_T}\boldsymbol{x}_0}{\sqrt{1-\alpha_T}} + \boldsymbol{\delta}_T^k, \sigma_T^2\boldsymbol{I} - \boldsymbol{\Delta}_T^k\right)$$

$$q_{\sigma,\mathcal{M}}(\boldsymbol{x}_T|\boldsymbol{x}_0)\,\mathrm{d}\boldsymbol{x}_T$$

$$= \sum_{k=1}^{K} \pi_T^k \int_{\boldsymbol{x}_T} \mathcal{N}\left(\sqrt{\alpha_{T-1}}\boldsymbol{x}_0 + \sqrt{1-\alpha_{T-1}-\sigma_T^2}.\frac{\boldsymbol{x}_T - \sqrt{\alpha_T}\boldsymbol{x}_0}{\sqrt{1-\alpha_T}} + \boldsymbol{\delta}_T^k, \sigma_T^2\boldsymbol{I} - \boldsymbol{\Delta}_T^k\right)$$

$$\mathcal{N}\left(\sqrt{\alpha_T}\boldsymbol{x}_0, (1-\alpha_T)\boldsymbol{I}\right)\,\mathrm{d}\boldsymbol{x}_T,$$

$$= \sum_{k=1}^{K} \pi_T^k \mathcal{N}\left(\sqrt{\alpha_{T-1}}\boldsymbol{x}_0 + \boldsymbol{\delta}_T^k, (1-\alpha_{T-1})\boldsymbol{I} - \boldsymbol{\Delta}_T^k\right), \tag{23}$$

which is also a GMM with the same mixing weights $\pi_T^k$. This is due to the fact that each of the above integrals is a Gaussian, whose parameters can be determined by using Gaussian marginalization identities (Bishop, 2006)(2.115). The mean $\boldsymbol{\mu}_{T-1}^{GMM}$ and the covariance $\boldsymbol{\Sigma}_{T-1}^{GMM}$ parameters of the above GMM are given by

$$\boldsymbol{\mu}_{T-1}^{GMM} = \sum_{k=1}^{K} \pi_T^k \left(\sqrt{\alpha_{T-1}}\boldsymbol{x}_0 + \boldsymbol{\delta}_T^k\right)$$

$$= \sqrt{\alpha_{T-1}}\boldsymbol{x}_0 + \sum_{k=1}^{K} \pi_T^k \boldsymbol{\delta}_T^k$$

$$\boldsymbol{\Sigma}_{T-1}^{GMM} = \sum_{k=1}^{K} \pi_T^k \left((1-\alpha_{T-1})\boldsymbol{I} - \boldsymbol{\Delta}_T^k\right) + \sum_{k=1}^{K} \pi_T^k (\boldsymbol{\delta}_T^k - \bar{\boldsymbol{\delta}}_T)(\boldsymbol{\delta}_T^k - \bar{\boldsymbol{\delta}}_T)^T$$

$$= (1-\alpha_{T-1})\boldsymbol{I} + \sum_{k=1}^{K} \pi_T^k \left((\boldsymbol{\delta}_T^k - \bar{\boldsymbol{\delta}}_T)(\boldsymbol{\delta}_T^k - \bar{\boldsymbol{\delta}}_T)^T - \boldsymbol{\Delta}_T^k\right), \tag{24}$$

where $\bar{\boldsymbol{\delta}}_T = \sum_{k=1}^{K} \pi_T^k \boldsymbol{\delta}_T^k$. It is straightforward to verify that these are equal to the desired means and covariance parameters of the equivalent DDPM forward marginal if the constraints in Eq. 8 are satisfied. Specifically the constraint $\bar{\boldsymbol{\delta}}_T = 0$ and either one of the constraints on $\boldsymbol{\Delta}_T^k$ in Eq. 8 lead to the following expressions for the first and second order moments:

$$\boldsymbol{\mu}_{T-1}^{GMM} = \sqrt{\alpha_{T-1}}\boldsymbol{x}_0$$
$$\boldsymbol{\Sigma}_{T-1}^{GMM} = (1-\alpha_{T-1})\boldsymbol{I}, \tag{25}$$

as desired.

The marginal of $\boldsymbol{x}_{T-2}$ can be derived similarly by invoking Bayes' rule and using the form of the GMM for $\boldsymbol{x}_{T-1}$. Specifically

$$q_{\sigma,\mathcal{M}}(\boldsymbol{x}_{T-2}|\boldsymbol{x}_0) = \int_{\boldsymbol{x}_{T-1}} q_{\sigma,\mathcal{M}}(\boldsymbol{x}_{T-2}|\boldsymbol{x}_{T-1},\boldsymbol{x}_0) q_{\sigma,\mathcal{M}}(\boldsymbol{x}_{T-1}|\boldsymbol{x}_0)\,\mathrm{d}\boldsymbol{x}_{T-1}$$

$$= \int_{\boldsymbol{x}_{T-1}} \sum_{l=1}^{L} \pi_{T-1}^l \mathcal{N}\left(\sqrt{\alpha_{T-2}}\boldsymbol{x}_0 + \sqrt{1-\alpha_{T-2}-\sigma_{T-1}^2}\cdot\frac{\boldsymbol{x}_{T-1}-\sqrt{\alpha_{T-1}}\boldsymbol{x}_0}{\sqrt{1-\alpha_{T-1}}} + \boldsymbol{\delta}_{T-1}^l, \sigma_{T-1}^2\boldsymbol{I}-\boldsymbol{\Delta}_{T-1}^l\right)$$

$$q_{\sigma,\mathcal{M}}(\boldsymbol{x}_{T-1}|\boldsymbol{x}_0)\,\mathrm{d}\boldsymbol{x}_{T-1}$$

$$= \int_{\boldsymbol{x}_{T-1}} \left\{ \sum_{l=1}^{L} \pi_{T-1}^l \mathcal{N}\left(\sqrt{\alpha_{T-2}}\boldsymbol{x}_0 + \sqrt{1-\alpha_{T-2}-\sigma_{T-1}^2}\cdot\frac{\boldsymbol{x}_{T-1}-\sqrt{\alpha_{T-1}}\boldsymbol{x}_0}{\sqrt{1-\alpha_{T-1}}} + \boldsymbol{\delta}_{T-1}^l, \sigma_{T-1}^2\boldsymbol{I}-\boldsymbol{\Delta}_{T-1}^l\right) \right\}$$

$$\left\{ \sum_{k=1}^{K} \pi_T^k \mathcal{N}\left(\sqrt{\alpha_{T-1}}\boldsymbol{x}_0 + \boldsymbol{\delta}_T^k, (1-\alpha_{T-1})\boldsymbol{I}-\boldsymbol{\Delta}_T^k\right) \right\}\,\mathrm{d}\boldsymbol{x}_{T-1}$$

$$= \sum_{k=1}^{K}\sum_{l=1}^{L} \pi_T^k \pi_{T-1}^l \int_{\boldsymbol{x}_{T-1}} \mathcal{N}\left(\sqrt{\alpha_{T-2}}\boldsymbol{x}_0 + \sqrt{1-\alpha_{T-2}-\sigma_{T-1}^2}\cdot\frac{\boldsymbol{x}_{T-1}-\sqrt{\alpha_{T-1}}\boldsymbol{x}_0}{\sqrt{1-\alpha_{T-1}}} + \boldsymbol{\delta}_{T-1}^l, \sigma_{T-1}^2\boldsymbol{I}-\boldsymbol{\Delta}_{T-1}^l\right)$$

$$\mathcal{N}\left(\sqrt{\alpha_{T-1}}\boldsymbol{x}_0 + \boldsymbol{\delta}_T^k, (1-\alpha_{T-1})\boldsymbol{I}-\boldsymbol{\Delta}_T^k\right)\,\mathrm{d}\boldsymbol{x}_{T-1}$$

$$= \sum_{k=1}^{K}\sum_{l=1}^{L} \pi_T^k \pi_{T-1}^l \mathcal{N}\left(\boldsymbol{\mu}_{k,l}^{T,T-1}, \boldsymbol{\Sigma}_{k,l}^{T,T-1}\right), \tag{26}$$

where we assume that the transition kernel from $\boldsymbol{x}_{T-1}$ to $\boldsymbol{x}_{T-2}$ is a GMM with $L$ components and parameters $\left(\pi_{T-1}^l, \boldsymbol{\delta}_{T-1}^l, \boldsymbol{\Delta}_{T-1}^l\right), l = 1\ldots L$. Each of the integrals above is a Gaussian whose mean $\boldsymbol{\mu}_{k,l}^{T,T-1}$ and covariance $\boldsymbol{\Sigma}_{k,l}^{T,T-1}$ parameters can be deduced by invoking Gaussian marginalization identities (Bishop, 2006)(2.115) and are given by:

$$\boldsymbol{\mu}_{k,l}^{T,T-1} = \sqrt{\alpha_{T-2}}\boldsymbol{x}_0 + \frac{\sqrt{1-\alpha_{T-2}-\sigma_{T-1}^2}}{\sqrt{1-\alpha_{T-1}}}\boldsymbol{\delta}_T^k + \boldsymbol{\delta}_{T-1}^l$$

$$\boldsymbol{\Sigma}_{k,l}^{T,T-1} = (1-\alpha_{T-2})\boldsymbol{I} - \frac{1-\alpha_{T-2}-\sigma_{T-1}^2}{1-\alpha_{T-1}}\boldsymbol{\Delta}_T^k - \boldsymbol{\Delta}_{T-1}^l. \tag{27}$$

Using the above expressions for the mean and covariance parameters of individual Gaussian components, the corresponding parameters $\boldsymbol{\mu}_{T-2}^{GMM}$ and $\boldsymbol{\Sigma}_{T-2}^{GMM}$ for the GMM marginal of $\boldsymbol{x}_{T-2}$ are given by:

$$\boldsymbol{\mu}_{T-2}^{GMM} = \sum_{k=1}^{K}\sum_{l=1}^{L} \pi_T^k \pi_{T-1}^l \boldsymbol{\mu}_{k,l}^{T,T-1}$$

$$= \sqrt{\alpha_{T-2}}\boldsymbol{x}_0 \tag{28}$$

$$\tag{29}$$

$$\boldsymbol{\Sigma}_{T-2}^{GMM} = \sum_{k=1}^{K}\sum_{l=1}^{L} \pi_T^k \pi_{T-1}^l \boldsymbol{\Sigma}_{k,l}^{T,T-1} + \sum_{k=1}^{K}\sum_{l=1}^{L} \pi_T^k \pi_{T-1}^l [A\boldsymbol{\delta}_k^T + \boldsymbol{\delta}_l^{T-1}][A\boldsymbol{\delta}_k^T + \boldsymbol{\delta}_l^{T-1}]^T$$

$$= (1-\alpha_{T-2})\boldsymbol{I}, \tag{30}$$

where

$$A = \frac{\sqrt{1-\alpha_{T-2}-\sigma_{T-1}^2}}{\sqrt{1-\alpha_{T-1}}}. \tag{31}$$

We have made use of the constraints in Eq. 8, for the parameters $(\pi_k^T, \pi_l^{T-1}, \boldsymbol{\delta}_k^T, \boldsymbol{\delta}_l^{T-1}, \boldsymbol{\Delta}_k^T, \boldsymbol{\Delta}_l^{T-1})$, to arrive at the above expressions and noting that $A$ is independent of the GMM parameters $\mathcal{M}_T$ and $\mathcal{M}_{T-1}$. The first and second order moments in Eq. 29 and Eq. 30 correspond to the DDPM forward marginal moments of $\boldsymbol{x}_{T-2}$. The proof for all latents $\boldsymbol{x}_t, t < T - 2$ follows from a similar argument as above noting that the form of the marginal in Eq. 26 is a GMM with $M = KL$ components. Further each component's mean and covariances in Eq. 27 carry future $(T-1$ and $T)$ step transition kernels' offset parameters ($\boldsymbol{\delta}$'s and $\boldsymbol{\Delta}$'s) as linear additive factors with coefficients ($A$, $A^2$ and 1) that are independent of those parameters. This makes it easier to see why proof by induction should work for $t < T - 2$.

## A.2 Variance Upper Bounds

The upper bounds of the eigenvalues of the matrix $\boldsymbol{\Delta}_t^k$ are tractable because the matrix is a weighted sum of outer-products of mean centered orthonormal vectors. Specifically $\boldsymbol{\Delta}_t^k$ can be written as

$$\boldsymbol{\Delta}_t^k = \frac{s^2}{K\pi_t^k} \left( \sum_{l=1}^K \pi_t^k (\boldsymbol{u}_t^l)(\boldsymbol{u}_t^l)^T - \bar{\boldsymbol{u}}_t \bar{\boldsymbol{u}}_t^T \right), \tag{32}$$

where $\boldsymbol{u}_t^k = \boldsymbol{U}_t[k]$ (Section 3.1.2). Diagonalizing the first term in Eq. 32 by pre and post multiplying by the matrix $\boldsymbol{U}_t[1:K]$ leads to a matrix $\boldsymbol{M}_t^k$, which is a sum of a diagonal matrix and a rank one matrix,

$$\boldsymbol{M}_t^k = \boldsymbol{U}_t[1:K]^T \boldsymbol{\Delta}_t^k \boldsymbol{U}_t[1:K]$$
$$= \frac{s^2}{K\pi_t^k}(\boldsymbol{D}_t^k - \boldsymbol{\pi}_t \boldsymbol{\pi}_t^T), \tag{33}$$

where $\boldsymbol{D}_t^k$ is a diagonal matrix with $\pi_t^k, k = 1 \ldots K$, along its diagonal and $\boldsymbol{\pi}_t$ is a column vector of mixture proportions $\pi_t^k$. Using a bound (Golub, 1973) on the eigenvalues of a diagonal matrix modified by a rank one matrix, if $\lambda_i, i = 1 \ldots K$ are the eigenvalues of $\boldsymbol{M}_t^k$, then

$$\frac{s^2}{K\pi_t^k} \left( \pi_t^1 - \sum_{l=1}^K (\pi_t^l)^2 \right) \le \lambda_1 \le \frac{s^2}{K\pi_t^k} \pi_t^1$$
$$\frac{s^2}{K\pi_t^k} \pi_t^{i-1} \le \lambda_i \le \frac{s^2}{K\pi_t^k} \pi_t^i, \qquad i = 2 \ldots K. \tag{34}$$

## A.3 Forward Process

We can derive the forward process using Bayes' rule. Specifically

$$q_{\sigma,\mathcal{M}}(\boldsymbol{x}_t | \boldsymbol{x}_{t-1}, \boldsymbol{x}_0) = \frac{q_{\sigma,\mathcal{M}}(\boldsymbol{x}_{t-1} | \boldsymbol{x}_t, \boldsymbol{x}_0) q_{\sigma,\mathcal{M}}(\boldsymbol{x}_t | \boldsymbol{x}_0)}{q_{\sigma,\mathcal{M}}(\boldsymbol{x}_{t-1} | \boldsymbol{x}_0)}$$
$$= \frac{\sum_{k=1}^K \pi_t^k \mathcal{N}(\boldsymbol{\mu}_t^k, \boldsymbol{\Sigma}_t^k) \sum_{l=1}^L \pi_{t,marginal}^l \mathcal{N}(\boldsymbol{\mu}_{t,marginal}^l, \boldsymbol{\Sigma}_{t,marginal}^l)}{\sum_{m=1}^M \pi_{t-1,marginal}^m \mathcal{N}(\boldsymbol{\mu}_{t-1,marginal}^m, \boldsymbol{\Sigma}_{t-1,marginal}^m)},$$

where $(\pi_t^k, \boldsymbol{\mu}_t^k, \boldsymbol{\Sigma}_t^k), k = 1 \ldots K$ are the DDIM-GMM transition kernel parameters from step $t$ to $t-1$ as given by Eq. 7. The parameters $(\pi_{t,marginal}^*, \boldsymbol{\mu}_{t,marginal}^*, \boldsymbol{\Sigma}_{t,marginal}^*)$ correspond to the DDIM-GMM marginal at any step $t$, which is also a mixture of Gaussians as derived in Appendix A.1. Note that the inference process of the proposed implicit model is non-Gaussian and non-Markovian in general and different from Gaussian diffusion.

## A.4 Upper Bound of ELBO using the DDIM-GMM Inference Process

In this section we provide an upper bound for the ELBO loss using the proposed DDIM-GMM inference process in terms of an augmented version of the DDPM $\mathcal{L}_{simple,w}$ loss, w.r.t. the denoiser parameters $\theta$.

The ELBO loss $\mathcal{L}_{ELBO,q_{\sigma,\mathcal{M}}}(\theta)$ using the proposed DDIM-GMM inference process is given by

$$
\begin{aligned}
\mathcal{L}_{ELBO,q_{\sigma,\mathcal{M}}}(\theta) = \mathbb{E}_{q_{\sigma,\mathcal{M}}} & \left[ D_{KL}(q_{\sigma,\mathcal{M}}(\boldsymbol{x}_T|\boldsymbol{x}_0)||p_\theta(\boldsymbol{x}_T)) + \sum_{t=2}^{T} D_{KL}(q_{\sigma,\mathcal{M}}(\boldsymbol{x}_{t-1}|\boldsymbol{x}_t,\boldsymbol{x}_0)||p_\theta(\boldsymbol{x}_{t-1}|\boldsymbol{x}_t)) \right] \\
& - \mathbb{E}_{q_{\sigma,\mathcal{M}}} \left[ \log p_\theta(\boldsymbol{x}_0|\boldsymbol{x}_1) \right] \\
= \mathbb{E}_{q_{\sigma,\mathcal{M}}} & \left[ \sum_{t=2}^{T} D_{KL}(q_{\sigma,\mathcal{M}}(\boldsymbol{x}_{t-1}|\boldsymbol{x}_t,\boldsymbol{x}_0)||p_\theta(\boldsymbol{x}_{t-1}|\boldsymbol{x}_t)) - \log p_\theta(\boldsymbol{x}_0|\boldsymbol{x}_1) \right] + const.,
\end{aligned}
\tag{35}
$$

where *const.* is a term independent of $\theta$ because $p_\theta(\boldsymbol{x}_T) = \mathcal{N}(\boldsymbol{0}, \boldsymbol{I})$. We ignore the constant term and assume a normal likelihood function for the observation $\boldsymbol{x}_0$ given $\boldsymbol{x}_1$, i.e.,

$$
p_\theta(\boldsymbol{x}_0|\boldsymbol{x}_1) = \mathcal{N}(\boldsymbol{f}_\theta(\boldsymbol{x}_1, 1), \sigma_1^2 \boldsymbol{I}),
\tag{36}
$$

where $\boldsymbol{f}_\theta(\boldsymbol{x}_t, t)$ is the denoiser estimate of $\boldsymbol{x}_0$, given by:

$$
\boldsymbol{f}_\theta(\boldsymbol{x}_t, t) = \frac{\boldsymbol{x}_t - \sqrt{1 - \alpha_t}\boldsymbol{\epsilon}_\theta(\boldsymbol{x}_t, t)}{\sqrt{\alpha_t}}.
\tag{37}
$$

The ELBO loss in Eq. 35 reduces to

$$
\begin{aligned}
\mathcal{L}_{ELBO,q_{\sigma,\mathcal{M}}}(\theta) &= \sum_{t=2}^{T} \mathbb{E}_{q_{\sigma,\mathcal{M}_t}} \left[ D_{KL}(q_{\sigma,\mathcal{M}_t}(\boldsymbol{x}_{t-1}|\boldsymbol{x}_t,\boldsymbol{x}_0)||q_{\sigma,\mathcal{M}_t}(\boldsymbol{x}_{t-1}|\boldsymbol{x}_t,\boldsymbol{f}_\theta(\boldsymbol{x}_t,t)))) \right] \\
&\quad + \mathbb{E}_{q_{\sigma,\mathcal{M}}(\boldsymbol{x}_0,\boldsymbol{x}_1)} \left[ -\log p_\theta(\boldsymbol{x}_0|\boldsymbol{x}_1) \right] \\
&:= K_1 + K_2 \\
&= \sum_{t=2}^{T} K_{1,t} + K_2,
\end{aligned}
\tag{38}
$$

where we use $p_\theta(\boldsymbol{x}_{t-1}|\boldsymbol{x}_t) = q_{\sigma,\mathcal{M}}(\boldsymbol{x}_{t-1}|\boldsymbol{x}_t,\boldsymbol{f}_\theta(\boldsymbol{x}_t,t))$ as in DDIM (Song et al., 2021). The first term $K_1$ involves KL-Divergences between mixtures of Gaussians, which is analytically intractable. However, we can use a suitable upper bound (Hershey & Olsen, 2007) as a surrogate for optimization. Assuming that there is a one-to-one correspondence between the mixture components of the GMMs using the true and estimated value of $\boldsymbol{x}_0$ above, we can use the matched bound (Hershey & Olsen, 2007; Do, 2003) as the upper bound to each of the KLD terms $K_{1,t}$ at step $t$, i.e. for any $t > 1$

$$
\begin{aligned}
K_{1,t} &\leq \mathbb{E}_{q_{\sigma,\mathcal{M}}} \left[ \sum_k \pi_t^k D_{KL}((q_{\sigma,\mathcal{M}_t^k}(\boldsymbol{x}_{t-1}|\boldsymbol{x}_t,\boldsymbol{x}_0)||q_{\sigma,\mathcal{M}_t^k}(\boldsymbol{x}_{t-1}|\boldsymbol{x}_t,\boldsymbol{f}_\theta(\boldsymbol{x}_t,t))))) \right] \\
&\leq \mathbb{E}_{q(\boldsymbol{x}_0)q_{\sigma,\mathcal{M}}(\boldsymbol{x}_t|\boldsymbol{x}_0)\boldsymbol{\epsilon}\sim\mathcal{N}(\boldsymbol{0},\boldsymbol{I})} \left[ \left( \sum_k \frac{\pi_t^k}{\nu_t^k} \right) \frac{(1-\alpha_t)}{2\alpha_t} ||\boldsymbol{\epsilon} - \boldsymbol{\epsilon}_\theta(\boldsymbol{x}_t,t)||^2 \right] \\
&= \sum_l \xi_t^{GMM,l} \mathbb{E}_{q(\boldsymbol{x}_0)q_{\sigma,\mathcal{M}}^l(\boldsymbol{x}_t|\boldsymbol{x}_0)\boldsymbol{\epsilon}\sim\mathcal{N}(\boldsymbol{0},\boldsymbol{I})} \left[ \left( \sum_k \frac{\pi_t^k}{\nu_t^k} \right) \frac{(1-\alpha_t)}{2\alpha_t} ||\boldsymbol{\epsilon} - \boldsymbol{\epsilon}_\theta(\boldsymbol{x}_t,t)||^2 \right] \\
&= \mathbb{E}_{l\sim\xi_t^{GMM}q(\boldsymbol{x}_0)q_{\sigma,\mathcal{M}}^l(\boldsymbol{x}_t|\boldsymbol{x}_0)\boldsymbol{\epsilon}\sim\mathcal{N}(\boldsymbol{0},\boldsymbol{I})} \left[ \left( \sum_k \frac{\pi_t^k}{\nu_t^k} \right) \frac{(1-\alpha_t)}{2\alpha_t} ||\boldsymbol{\epsilon} - \boldsymbol{\epsilon}_\theta(\boldsymbol{x}_t,t)||^2 \right]
\end{aligned}
\tag{39}
$$

where $\nu_t^k$ is the minimum variance within a diagonal approximation of the covariance matrix $\sigma_t^2\boldsymbol{I} - \boldsymbol{\Delta}_t^k$, i.e., $\nu_t^k = \min diag((\sigma_t^2\boldsymbol{I} - diag\_approx(\boldsymbol{\Delta}_t^k)))$. Note that in the above, $q_{\sigma,\mathcal{M}_t^k}(\boldsymbol{x}_{t-1}|\boldsymbol{x}_t,\boldsymbol{x}_0)$ is used to refer to the $k$th mixture component's density function of the GMM transition kernel. Similarly, $q_{\sigma,\mathcal{M}}^l(\boldsymbol{x}_t|\boldsymbol{x}_0)$ refers to the $l$th component of the DDIM-GMM's forward marginal GMM at step $t$. The upper bound of Eq. 39 can be interpreted as an augmented form of $\mathcal{L}_{simple,w}$ with weights

$$
w_t = \left( \sum_k \frac{\pi_t^k}{\nu_t^k} \right) \frac{(1-\alpha_t)}{2\alpha_t}
\tag{40}
$$

and the DDPM marginals' mean and covariance randomly modified with shifts from one of the DDIM-GMM forward marginal's components at every step $t$ (e.g., Eq. 27 for $t = T - 2$). The choice of the shifts is according to a discrete distribution with proportions given by the DDIM-GMM marginal's mixture priors $\xi_t^{GMM}$ (e.g. Eq. 26 for $t = T - 2$).

For $t = 1$, the loss term $K_2$ is given by

$$
\begin{aligned}
K_2 &= \mathbb{E}_{q_{\sigma,\mathcal{M}}(\boldsymbol{x}_0,\boldsymbol{x}_1)} \left[ -\log p_\theta(\boldsymbol{x}_0|\boldsymbol{x}_1) \right] \\
&= \mathbb{E}_{q(\boldsymbol{x}_0)q_{\sigma,\mathcal{M}}(\boldsymbol{x}_1|\boldsymbol{x}_0)\boldsymbol{\epsilon}\sim\mathcal{N}(\boldsymbol{0},\boldsymbol{I})} \left[ \frac{(1-\alpha_1)}{2\sigma_1^2\alpha_1} ||\boldsymbol{\epsilon} - \boldsymbol{\epsilon}_\theta(\boldsymbol{x}_1,1)||^2 \right] + const. \\
&= \mathbb{E}_{l\sim\xi_1^{GMM}q(\boldsymbol{x}_0)q_{\sigma,\mathcal{M}}^l(\boldsymbol{x}_1|\boldsymbol{x}_0)\boldsymbol{\epsilon}\sim\mathcal{N}(\boldsymbol{0},\boldsymbol{I})} \left[ \frac{(1-\alpha_1)}{2\sigma_1^2\alpha_1} ||\boldsymbol{\epsilon} - \boldsymbol{\epsilon}_\theta(\boldsymbol{x}_1,1)||^2 \right],
\end{aligned}
\tag{41}
$$

where we have ignored the *const.* term independent of $\theta$. Combining Eqs. 39 and 41, the $\mathcal{L}_{ELBO,q_{\sigma,\mathcal{M}}}(\theta)$ can be interpreted as upper bounded by an augmented version of $\mathcal{L}_{simple,w}$ with weights $w_t$ given in Eqs. 40 and 41.

## A.5 Experimental Details

We provide additional details on the experiments reported in Section 5, specifically for the CelebAHQ, FFHQ and ImageNet experiments. All our diffusion models are trained in the latent space of a VQVAE (Rombach et al., 2022). The input images to the VQVAE are at a resolution of 256x256 pixels. Each of the VQVAEs are trained on a large scale dataset. Specifically the VQVAEs for unconditional generation on CelebAHQ and class-conditional generation on ImageNet are trained on OpenImages. We use the publicly available $f4$ VQVAE (Table 8, Section D.2. of Rombach et al. (2022)) for training CelebAHQ models and $f8$ VQVAE for class-conditional ImageNet models respectively. The $f4$ VQVAE (#embeddings=8192) does not use attention layers at any resolution within the model architecture, whereas the $f8$ VQVAE (#embeddings=16384) uses attention at resolution 32. We train a $f4$ VQVAE (#embeddings=8192), with no attention layers, on ImageNet for 712k steps and use its latent space to train the diffusion models on FFHQ.

All our diffusion models are trained with 1000 forward steps using a linear noise ($\beta_t = 1 - \frac{\alpha_t}{\alpha_{t-1}}$) schedule of $[\beta_0 = 0.0015, \beta_{1000} = 0.0195]$. We use the U-Net architecture (Ho et al., 2020; Rombach et al., 2022) for the denoiser. Specifically, the unconditional U-Net encoders operating on the $f4$ VQVAE latent space have four 2x downsampling levels with channel multiplication factors of $[1, 2, 3, 4]$ starting from a base set of 224 channels. Each level uses two residual blocks. Attention blocks are used within levels at downsampling factors $[2, 4, 8]$. Similarly, the class-conditional U-Net encoders operating on $f8$ VQVAE latent space have three 2x downsampling levels with channel multiplication factors of $[1, 2, 4]$ starting from a base set of 256 channels. Other architectural details remain the same as before except that the attention blocks are at downsampling levels $[1, 2, 4]$.

For each DDIM-GMM variant, we choose a GMM with 8 mixture components with uniform priors ($\pi_t^k=0.125$) for all steps $t$. We also search for the best value of scaling $s$ among $\{0.01, 0.1, 1.0, 10.0\}$ and report the best result with the chosen value for $s$. It is possible that for some choices of $s$, the diagonal elements of $\boldsymbol{\Delta}_k^t$ or the corresponding upper bounds in the DDIM-GMM-VUB sampler could be larger than $\sigma_t^2$. In such cases we clip the negative elements of $(\sigma_t^2\boldsymbol{I} - diag\_approx(\boldsymbol{\Delta}_t^k))$ to zero, which amounts to sampling with zero variances in those dimensions in the latent space.

## A.6 Class-conditional ImageNet with Classifier Guidance

For classifier guidance, we train a separate classifier at different levels of noise and use it with two guidance scales (1, 10) for sampling with 10 and 100 steps. The FID and IS results are in Figs. 5 and 6 respectively. Using smaller guiance scale (1), DDIM-GMM-* samplers show improvements over DDIM only under the highest $\eta(= 1)$ setting. FID improves using the fewest sampling steps (10) and IS improves using both 10 and 100 sampling steps. This can be attributed to a similar argument as for unconditional sampling, especially for the least number of sampling steps. With a higher guidance scale (10), all variants of DDIM-GMM-* samplers yield significantly lower FIDs than the DDIM sampler when the number of sampling steps

is small (10) (see Table 10, Appendix A.14.3). The FIDs with variance upper bounding, relative to without, improve significantly with higher values of $\eta$ possibly due to greater exploration of the latent space under those settings. The differences between DDIM and DDIM-GMM-* are marginal using 100 sampling steps with the exception of DDIM-GMM-RAND and DDIM-GMM-ORTHO for the highest $\eta$ setting. With a higher guidance scale (10), the IS scores of samples from DDIM-GMM-* samplers are almost always higher than from the DDIM sampler (see Table 11, Appendix A.14.3). The only exception is the DDIM-GMM-ORTHO sampler run for 100 steps using $\eta = 1$, which is only marginally worse. Similar to FID results, the differences between samplers with and without variance upper bounding are amplified by $\eta$. This is an interesting result indicating that better exploration of latent spaces with a multimodal reverse kernel not only helps with coverage (FID) but also sample sharpness (IS) since the guidance scale is known to trade-off one versus the other (Dhariwal & Nichol, 2021) and poses challenges for higher order ODE solvers (Lu et al., 2023).

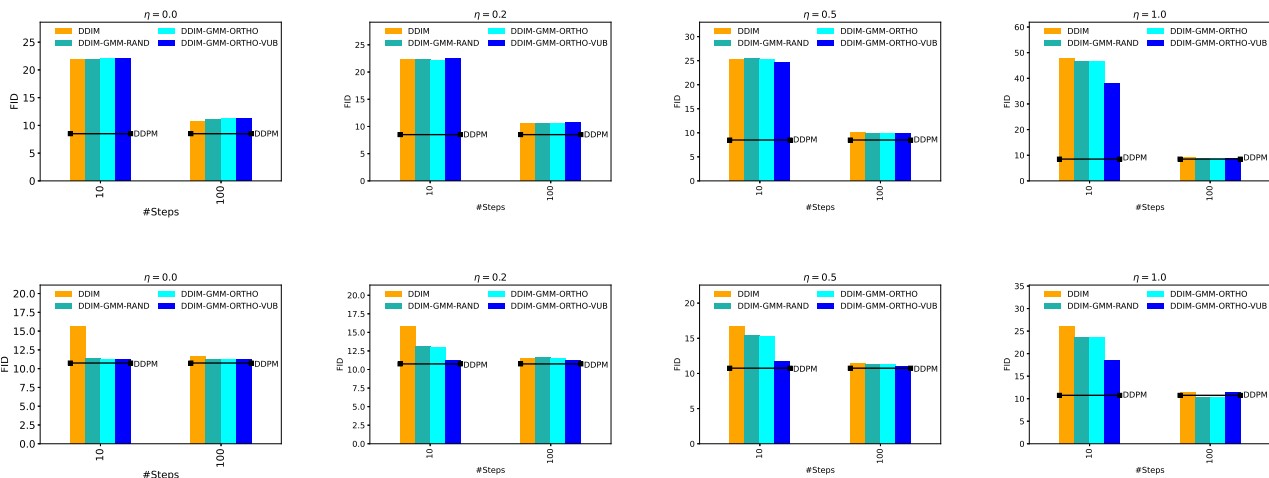

Figure 5: **Class-conditional ImageNet with Classifier Guidance**. FID ($\downarrow$) with guidance scale of 1.0 (top) and 10.0 (bottom) respectively. The horizontal line is the DDPM baseline run for 1000 steps.

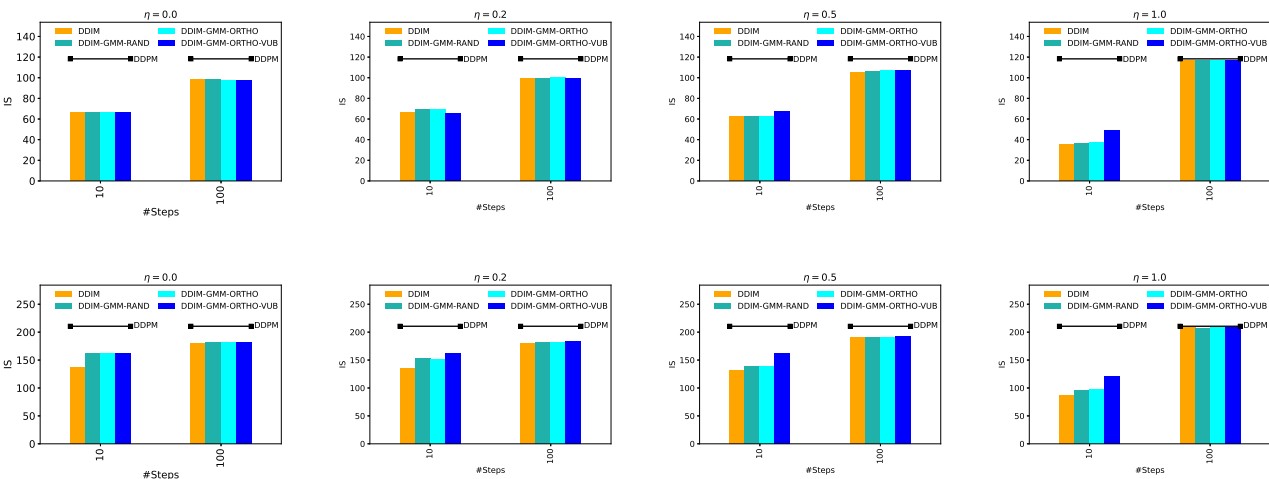

Figure 6: **Class-conditional ImageNet with Classifier Guidance**. IS ($\uparrow$) with guidance scale of 1.0 (top) and 10.0 (bottom) respectively. The horizontal line is the DDPM baseline run for 1000 steps.

### A.7 Class-conditional ImageNet with Classifier-free Guidance

In addition to the results reported in Section 5.2, we conduct experiments using a higher classifier-free guidance scale of 5.0. The FID and IS metrics are shown in Fig. 7. As expected, higher guidance leads to better IS values at the expense of FID compared to lower guidance. Using 10 sampling steps, the best IS of 320.66 (Table 13) is obtained with a deterministic ($\eta = 0$) DDIM-GMM-ORTHO sampler. Similarly, using 100 sampling steps, DDIM-GMM-ORTHO and DDIM-GMM-RAND yield the best IS, around 355, at $\eta = 1$. The FIDs are generally worse relative to sampling with lower guidance scale with the best value of 15.73 achieved using both DDIM-GMM-RAND and DDIM-GMM-ORTHO variants at $\eta = 0.5$.

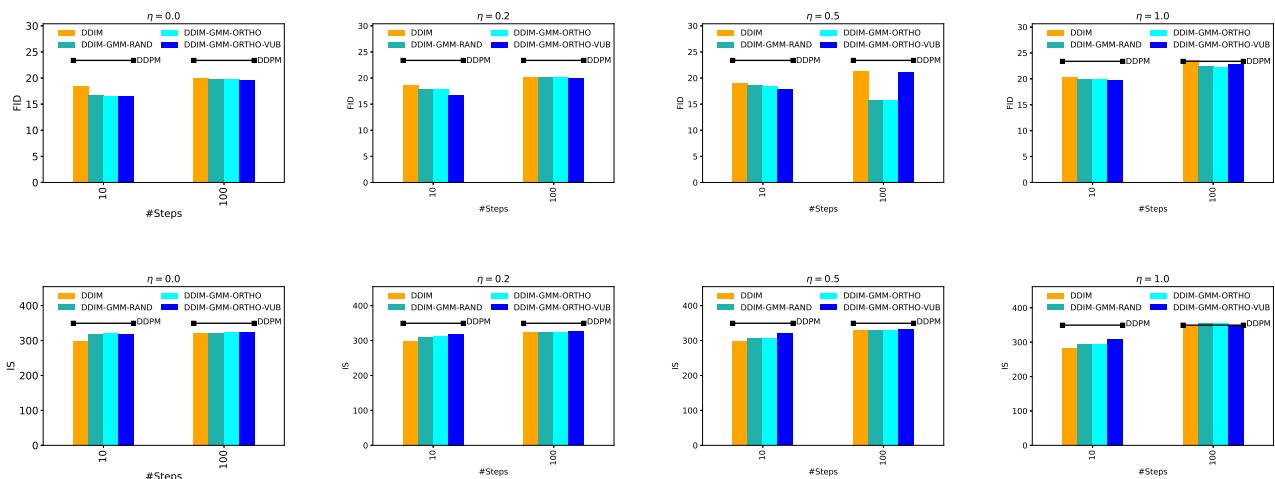

Figure 7: **Class-conditional ImageNet with Classifier-free Guidance**. FID ($\downarrow$) (top) and IS ($\uparrow$) (bottom) respectively with a guidance scale of 5.0. The horizontal line is the DDPM baseline run for 1000 steps.

### A.8 Ablations

In this section, we perform an ablative study on the number of mixture components and offset scaling factor $s$ of the GMM parameters using the unconditional model trained on the CelebAHQ dataset as described in Section 5. We conduct ablations for all three DDIM-GMM variants: RAND, ORTHO, and ORTHO-VUB, and fix the mixture weights of the components to be uniform in all experiments.

#### A.8.1 Number of mixture components

We compute the FID on the validation set by choosing one of 8, 256, or 1024 components ($K$) at each step during sampling, using different values of $\eta$. For each choice of $K$, we select the scale $s$ among (0.01, 0.1, 1.0, 10.0, 20.0) that leads to the lowest FID. The results are plotted in Fig. 8 for DDIM-GMM-RAND (top), DDIM-GMM-ORTHO (middle), and DDIM-GMM-ORTHO-VUB (bottom). For lower values of $\eta$ (0, 0.2), the performance is similar across all choices of $K$ for all three variants, since the offsets perturb only the means and the offset variances ($diag\_approx(\sigma_t^2 \boldsymbol{I} - \boldsymbol{\Delta}_t^k)$) are small. At higher $\eta$ values (0.5, 1.0), different choices of $K$ lead to similar results for DDIM-GMM-RAND and DDIM-GMM-ORTHO. For DDIM-GMM-ORTHO-VUB, $K = 8$ and $K = 256$ perform slightly better than $K = 1024$, since the latter restricts variances in too many dimensions, impacting exploration. As the number of steps increases, all choices converge to similar FID values across all variants and $\eta$ settings.

#### A.8.2 Offset scaling $s$

In order to study the effect of $s$, we fix the number of mixture components to $K = 8$ and choose a value for $s$ within five choices: (0.01, 0.1, 1.0, 10.0, 20.0). The results are shown in Fig. 9 for DDIM-GMM-RAND

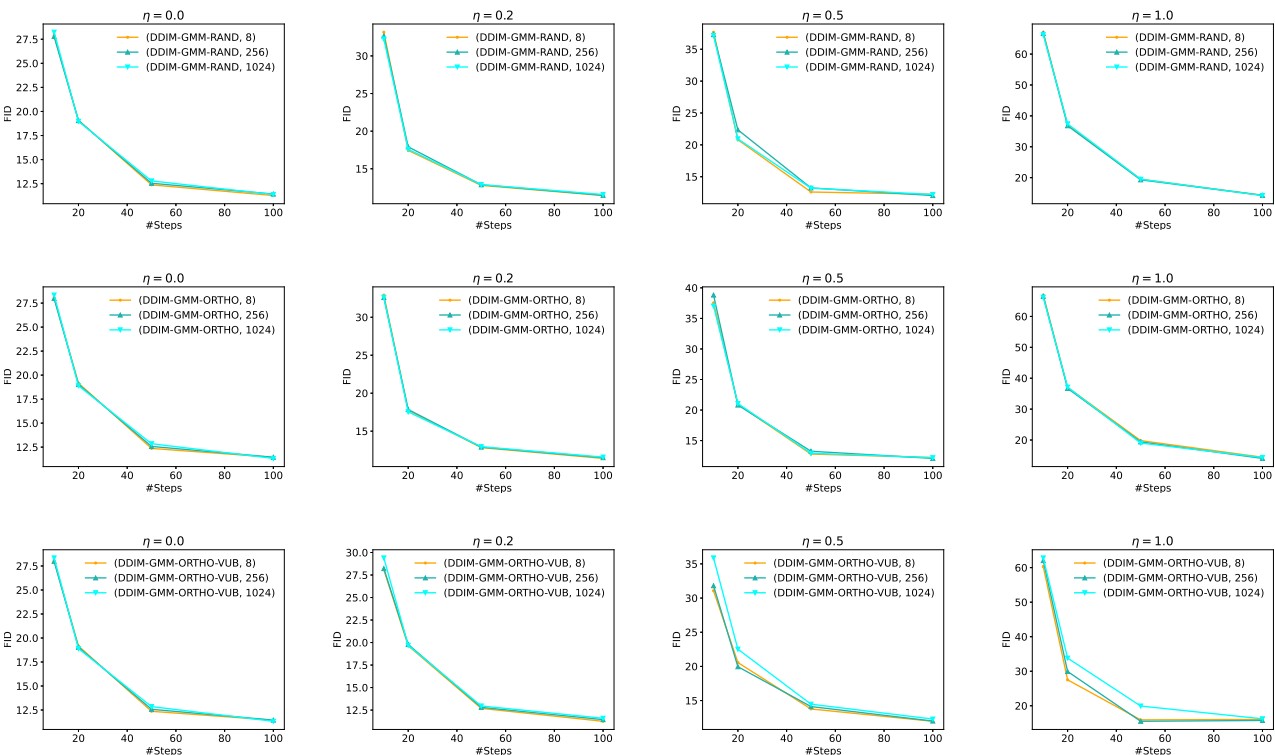

Figure 8: **CelebAHQ**. FID ($\downarrow$). Ablations on the number of mixture components for DDIM-GMM-RAND (top), DDIM-GMM-ORTHO (middle), and DDIM-GMM-ORTHO-VUB (bottom).

(top), DDIM-GMM-ORTHO (middle), and DDIM-GMM-ORTHO-VUB (bottom). The sample quality is almost the same with smaller values of $s$ (0.01–1.0) across all variants. For DDIM-GMM-RAND and DDIM-GMM-ORTHO, the highest value $s = 20$ leads to the best performance for up to 20 and 50 sampling steps at the larger $\eta$ values of 0.5 and 1.0 respectively, consistent with the hypothesis (Guo et al., 2023; Xiao et al., 2022) that true denoising distributions are multimodal at fewer sampling steps and larger exploration (higher $s$) is favorable. For DDIM-GMM-ORTHO-VUB, $s = 20$ yields the best performance only at $\eta = 1.0$ with 10 sampling steps, while $s = 10$ gives the best FID for fewer steps (10) at small $\eta$ (0.0, 0.2) and for up to 50 steps at $\eta = 1.0$. This can be explained by the interaction between $\eta$ ($\sigma_t^2$) and variance upper bounds (Appendix A.2) leading to lower variances for smaller $\eta$ and higher offset magnitudes at the same time in the case of DDIM-GMM-ORTHO-VUB. In all other cases, larger offsets lead to instability.

## A.9   Sharing GMM Parameters across Sampling Steps

As discussed in Section 3.1.3, we also experiment with sharing GMM parameters $\mathcal{M}_t$ across sampling steps $t$ by choosing the offsets only once followed by orthogonalization (SVD), scaling and variance upper bounding. This saves some compute time during initialization. Table 3 compares the FIDs between DDIM-GMM-ORTHO-VUB samplers that share GMM parameters (ORTHO-VUB*) with the corresponding ones that set them independently (ORTHO-VUB) across sampling steps $t$. We observe that there is no significant (more than 1 FID point) impact on sample FIDs across all datasets, except with FFHQ with 10 sampling steps using $\eta = 1$, where the shared parameter sampler yields slightly better results. Here we show results with unconditional models on CelebAHQ and FFHQ. More results can be found in Tables 8 to 13 in the Appendix.

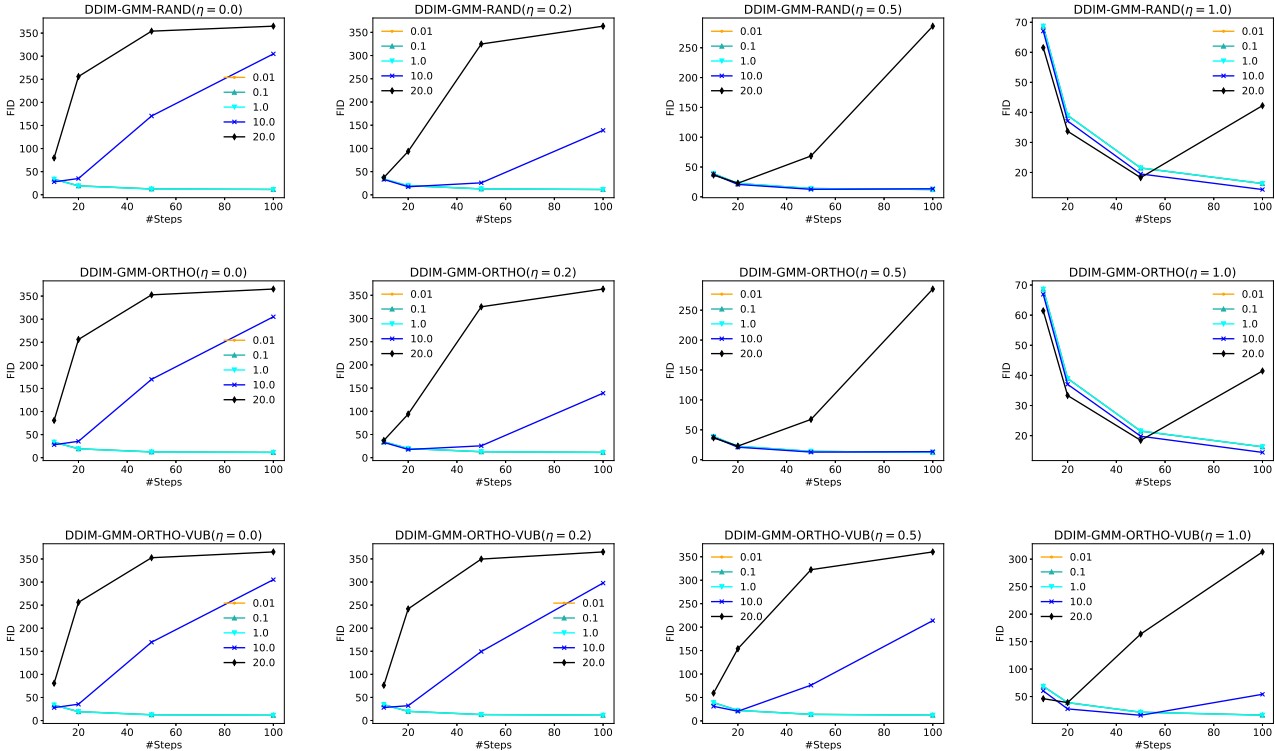

Figure 9: **CelebAHQ**. FID ($\downarrow$). Ablations on the offset scaling factor $s$ for DDIM-GMM-RAND (top), DDIM-GMM-ORTHO (middle), and DDIM-GMM-ORTHO-VUB (bottom).

Table 3: **Sharing GMM Parameters across Steps**. FID ($\downarrow$)

| DATASET | | CelebAHQ | | FFHQ | |
|---|---|---|---|---|---|
| STEPS | | 10 | 100 | 10 | 100 |
| $\eta$ | | | | | |
| 0 | ORTHO-VUB | 27.94 | 11.44 | 26.46 | 11.12 |
| 0 | ORTHO-VUB* | 27.84 | 11.42 | 27.18 | 11.24 |
| 1.0 | ORTHO-VUB | 60.65 | 15.60 | 69.25 | 11.44 |
| 1.0 | ORTHO-VUB* | 61.15 | 15.94 | **67.72** | 11.38 |

## A.10   Computational Overhead

As discussed briefly in Section 3.1, the proposed approach introduces additional computational overhead in an initialization phase prior to sampling. All the GMM mean and variance offsets are precomputed and saved in memory before sampling. We can choose to precompute a single set of GMM parameters per batch or the entire sample set. We experimented with both the options and did not see a significant difference in metrics. So it is computationally more efficient to precompute offsets once and fix them. We also experimented with choosing different GMM parameters for different sampling steps $t$ and found no significant difference with setting them the same across all $t$ in our experiments. Due to storing the additional GMM parameters, there is some memory overhead relative to DDIM but it is negligible, especially in the scenario of choosing a single set of offsets across all samples and time steps. In the scenario of using different offsets across subsets (batches) of samples or sampling steps, the overhead scales linearly along the sample subset size and number of step dimensions. The dimensionality of the latent spaces $\boldsymbol{x}_t$ also influence the computational and memory requirements of the GMM offset parameters. For instance, it might be infeasible to compute the outer products of centered offsets (Eq. 8) if the dimensionality of the latent spaces is high, e.g. high-resolution

image space diffusion models. In such cases, the DDIM-GMM-ORTHO-VUB sampler is more feasible as it provides an upper bound for the variance offsets without explicitly computing them.

To quantify the overhead in practice, we measured the average wall clock time per sample using the unconditional CelebAHQ model on a single NVIDIA Tesla V100 (32GB) GPU, running each sampler for 10 sampling steps with $K = 8$ mixture components, averaged over 100 samples.

| Sampler | Initialization (ms) | Sampling (ms/sample) |
|---|---|---|
| DDIM | — | 261 |
| DDIM-GMM-RAND | 0.9 | 281 |
| DDIM-GMM-ORTHO | 590 | 276 |
| DDIM-GMM-ORTHO-VUB | 590 | 268 |

Table 4: Wall clock times for DDIM and DDIM-GMM-* samplers on CelebAHQ (V100 GPU, 10 steps, $K = 8$).

The per-sample sampling overhead of the DDIM-GMM-* samplers relative to DDIM is small (3–8%). DDIM-GMM-RAND and DDIM-GMM-ORTHO incur slightly higher sampling cost than DDIM-GMM-ORTHO-VUB due to explicitly computing the full covariance matrix (Eq. 8), which DDIM-GMM-ORTHO-VUB avoids by using variance upper bounds. The initialization overhead for DDIM-GMM-ORTHO and DDIM-GMM-ORTHO-VUB (590ms) is dominated by the SVD operation and is incurred only once before sampling begins; DDIM-GMM-RAND requires negligible initialization (0.9ms) as no SVD is needed.

## A.11 Comparison with DPM-Solver

In this section, we compare DDIM-GMM-ORTHO-VUB and DPM-Solver (Lu et al., 2022) samplers on the class-conditional model trained on ImageNet. During inference, we use classifier-free guidance with weights (2.5, 5) and run each sampler for 10 and 100 steps. For the DDIM-GMM-ORTHO-VUB sampler, $\eta$ is set to 0. The results listed in Table 5 suggest DDIM-GMM-ORTHO-VUB is superior to DPM-Solver in all cases on the FID metric. This is also true with the IS metric with the exception of lower guidance scale (2.5) using 10 sampling steps.

| Steps | 10 | | 100 | |
|---|---|---|---|---|
| Guidance Scale | 2.5 | 5 | 2.5 | 5 |
| DPM-Solver | 9.75/**225.68** | 19.22/309.14 | 9.30/239.11 | 19.82/**323.87** |
| DDIM-GMM-ORTHO-VUB | **6.94**/207.85 | **16.61**/**319.13** | **9.13**/**240.88** | **19.66**/323.84 |

Table 5: Comparison with DPM-Solver. Class-conditional ImageNet with classifier free guidance (FID↓ / IS↑).

## A.12 Additional Experiments

In this section, we report results on additional experiments on the LSUN benchmarks (Yu et al., 2015) using the same settings as the DDIM (Song et al., 2021) work. Specifically, we use the pretrained DDPM models (Ho et al., 2020) on LSUN Bedroom and Church datasets to compare DDIM vs. DDIM-GMM-ORTHO-VUB samplers for different number of sampling steps (10, 20, 50 and 100).

| Steps | 10 | 20 | 50 | 100 |
|---|---|---|---|---|
| DDIM | 16.93 | 8.77 | 6.68 | 6.76 |
| DDIM-GMM | **16.86** | **8.76** | **6.62** | **6.67** |

Table 6: LSUN Bedroom. Comparison between DDIM and DDIM-GMM on the FID(↓) metric. $\eta = 0$ for both samplers.

| Steps | 10 | 20 | 50 | 100 |
|---|---|---|---|---|
| DDIM | 19.39 | **12.33** | 11.04 | 10.85 |
| DDIM-GMM | **19.33** | 12.37 | **10.85** | **10.81** |

Table 7: LSUN Church. Comparison between DDIM and DDIM-GMM on the FID($\downarrow$) metric. $\eta = 0$ for both samplers.

### A.13 Qualitative Results

In this section we show some qualitative results of sampling with the proposed approach compared to original DDIM. Specifically, we use the DDIM-GMM-ORTHO-VUB$^*$ (Section 3.1.3) method to obtain the samples and refer to them with the label DDIM-GMM for brevity. In Fig. 10 we plot samples from the classifier-guided class conditional model trained on ImageNet with 10 sampling steps for both DDIM (left) and DDIM-GMM (right). The top and bottom group of images correspond to input class labels "pelican" and "cairn terrier" respectively. See Fig. 11 for comparisons using classifier-free guidance. Also Fig. 12 shows images generated by conditioning on text prompts using the Stable Diffusion v2.1 model.

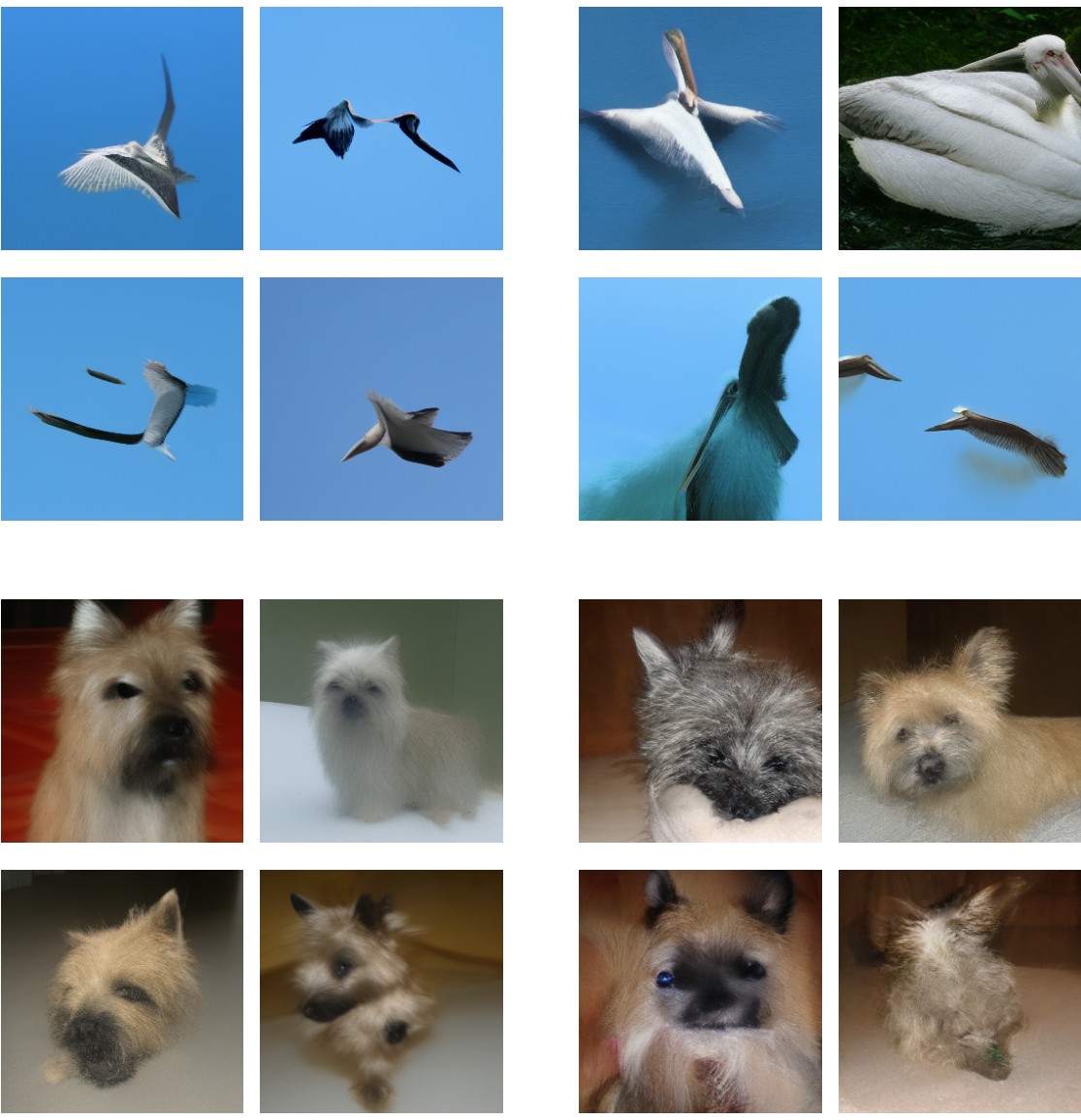

Figure 10: **Class-conditional ImageNet with classifier guidance, 10 sampling steps.** Random samples from the class-conditional ImageNet model using DDIM (left) and DDIM-GMM (right) sampler conditioned on the class labels *pelican* (top) and *cairn terrier* (bottom) respectively. 10 sampling steps are used for each sampler with a classifier guidance weight of 10 ($\eta = 1$).

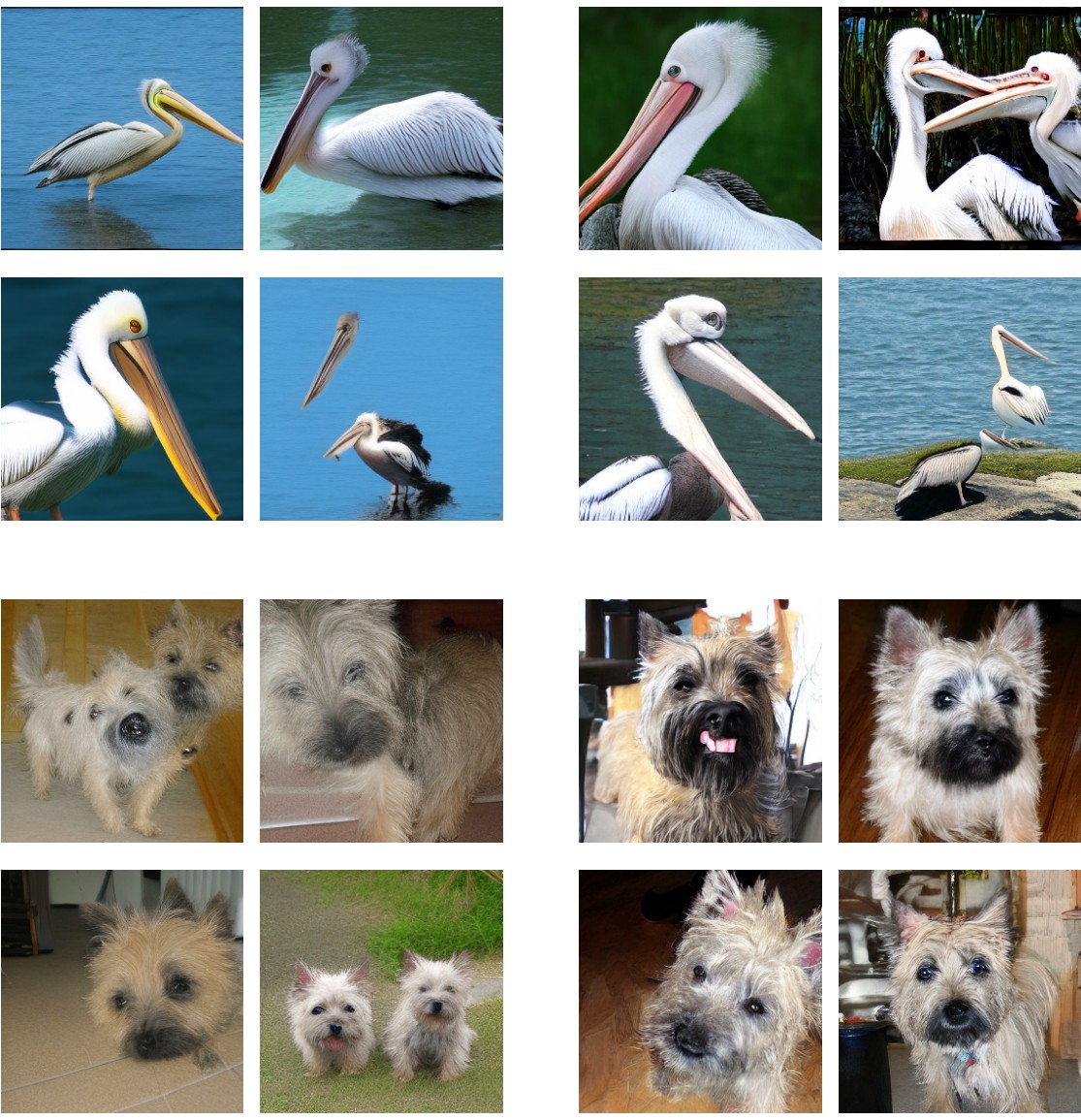

Figure 11: **Class-conditional ImageNet with classifier-free guidance, 10 sampling steps.** Random samples from the class-conditional ImageNet model using DDIM (left) and DDIM-GMM (right) sampler conditioned on the class labels *pelican* (top) and *cairn terrier* (bottom) respectively. 10 sampling steps are used for each sampler with a classifier free guidance weight of 5 ($\eta = 0$).

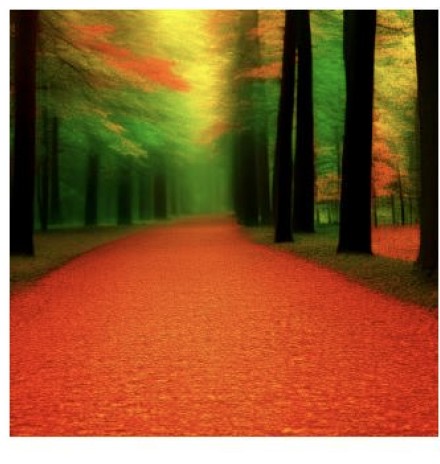
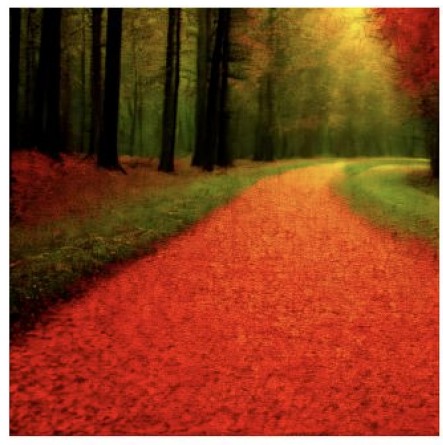

Autumn Forest Calmness Serenity Walk
Path Trees Nature

Autumn Forest Calmness Serenity Walk
Path Trees Nature

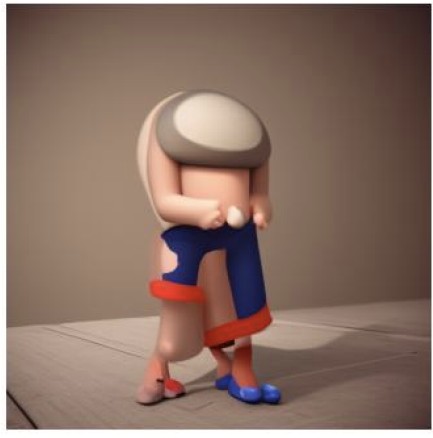
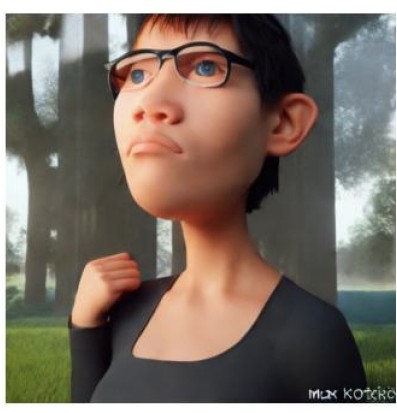

3d characer by max kostenko 23

3d characer by max kostenko 23

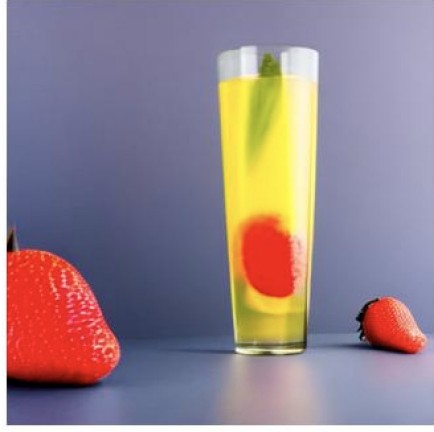
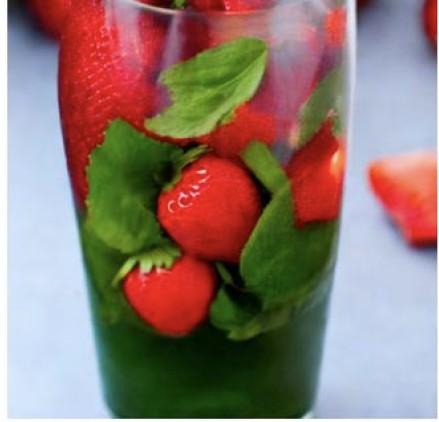

A tall glass with whole strawberries

A tall glass with whole strawberries

Figure 12: **Text-to-image-generation using Stable Diffusion v2.1, 10 sampling steps.** Random samples from the Stable Diffusion v2.1 model using DDIM (left) and DDIM-GMM (right) sampler conditioned on text prompts, displayed below each image. 10 sampling steps are used for each sampler with a classifier free guidance weight of 7.5 ($\eta = 0$).

### A.14 FID and IS Metrics

In this section we list the metrics plotted in Section 5 in a tabular format. The numbers in bold emphasize improvement of the corresponding sampling method's metric if the difference in metric (FID or IS) is at least 1 unit from the worst result within the same group (same $\eta$ and number of sampling steps). The number in parentheses denotes the scale parameter $s$ that resulted in the best metric for the particular DDIM-GMM-* sampling method under a given setting. We omit the best $s$ for ImageNet results.

### A.14.1 CelebAHQ

| | Steps | 10 | 20 | 50 | 100 | 1000 |
|---|---|---|---|---|---|---|
| $\eta$ | | | | | | |
| 0 | DDIM | 32.95 | 18.58 | 12.65 | 11.42 | |
| 0 | DDIM-GMM-RAND | **28.01 (10)** | 18.74 (1) | 12.33 (1) | 11.35 (1) | |
| 0 | DDIM-GMM-ORTHO | **27.97 (10)** | 18.71 (1) | 12.41 (1) | 11.44 (1) | |
| 0 | DDIM-GMM-ORTHO-VUB | **27.94 (10)** | 18.71 (1) | 12.41 (1) | 11.44 (1) | |
| 0 | DDIM-GMM-ORTHO-VUB$^*$ | **27.84 (10)** | 18.53 (1) | 12.62 (1) | 11.42 (0.1) | |
| 0.2 | DDIM | 33.74 | 19.48 | 12.79 | 11.41 | |
| 0.2 | DDIM-GMM-RAND | **32.32 (10)** | **17.26 (10)** | 12.80 (0.01) | 11.36 (0.1) | |
| 0.2 | DDIM-GMM-ORTHO | **32.42 (10)** | **17.33 (10)** | 12.79 (1) | 11.37 (0.01) | |
| 0.2 | DDIM-GMM-ORTHO-VUB | **28.32 (10)** | 19.18 (1) | 12.58 (1) | 11.29 (1) | |
| 0.2 | DDIM-GMM-ORTHO-VUB$^*$ | **27.68 (10)** | 19.77 (0.01) | 12.81 (1) | 11.34 (1) | |
| 0.5 | DDIM | 39.04 | 22.01 | 13.99 | 12.05 | |
| 0.5 | DDIM-GMM-RAND | **37.15 (10)** | **20.66 (10)** | **12.95 (10)** | 12.26 (1) | |
| 0.5 | DDIM-GMM-ORTHO | **37.27 (10)** | **20.78 (10)** | **12.98 (10)** | 12.25 (0.1) | |
| 0.5 | DDIM-GMM-ORTHO-VUB | **31.00 (10)** | **20.65 (10)** | 13.87 (1) | 12.00 (1) | |
| 0.5 | DDIM-GMM-ORTHO-VUB$^*$ | **31.42 (10)** | **20.89 (10)** | 14.09 (1) | 11.91 (1) | |
| 1.0 | DDIM | 68.67 | 39.20 | 21.53 | 16.09 | |
| 1.0 | DDIM-GMM-RAND | **66.94 (10)** | **37.63 (10)** | **19.33 (10)** | **14.14 (10)** | |
| 1.0 | DDIM-GMM-ORTHO | **67.15 (10)** | **37.79(1)** | **19.36 (10)** | **14.37 (10)** | |
| 1.0 | DDIM-GMM-ORTHO-VUB | **60.65 (10)** | **28.03 (10)** | **15.97 (10)** | 15.60 (1) | |
| 1.0 | DDIM-GMM-ORTHO-VUB$^*$ | **61.15 (10)** | **27.41 (10)** | **16.48 (10)** | 15.94 (1) | |
| 1.0 | DDPM | | | | | 11.59 |

Table 8: CelebAHQ (FID↓)

### A.14.2 FFHQ

| | Steps | 10 | 20 | 50 | 100 | 1000 |
|---|---|---|---|---|---|---|
| $\eta$ | | | | | | |
| 0 | DDIM | 28.73 | 15.68 | 11.67 | 11.17 | |
| 0 | DDIM-GMM-RAND | **26.55 (10)** | 15.64 (1) | 11.83 (0.01) | 11.12 (0.01) | |
| 0 | DDIM-GMM-ORTHO | **26.46 (10)** | 15.67 (1) | 11.83 (0.01) | 11.12 (0.01) | |
| 0 | DDIM-GMM-ORTHO-VUB | **26.46 (10)** | 15.67 (1) | 11.83 (0.01) | 11.12 (0.01) | |
| 0 | DDIM-GMM-ORTHO-VUB$^*$ | **27.18 (10)** | 15.43 (1) | 11.77 (0.1) | 11.24 (0.1) | |
| 0.2 | DDIM | 29.33 | 15.83 | 11.66 | 11.07 | |
| 0.2 | DDIM-GMM-RAND | 29.27 (0.1) | 16.03 (0.1) | 11.87 (0.01) | 10.92 (0.01) | |
| 0.2 | DDIM-GMM-ORTHO | 29.09 (10) | 16.01 (1) | 11.87 (0.01) | 10.92 (0.01) | |
| 0.2 | DDIM-GMM-ORTHO-VUB | **26.90 (10)** | 15.96 (1) | 11.87 (0.01) | 10.91 (0.1) | |
| 0.2 | DDIM-GMM-ORTHO-VUB$^*$ | **27.35 (10)** | 15.85 (0.01) | 11.62 (0.01) | 11.11 (0.01) | |
| 0.5 | DDIM | 35.53 | 17.83 | 11.89 | 10.53 | |
| 0.5 | DDIM-GMM-RAND | 35.16 (10) | 18.01 (10) | 11.85 (1) | 10.49 (0.01) | |
| 0.5 | DDIM-GMM-ORTHO | 35.00 (10) | 17.92 (10) | 11.87 (0.1) | 10.47 (0.1) | |
| 0.5 | DDIM-GMM-ORTHO-VUB | **29.03 (10)** | 18.16 (1) | 11.78 (1) | 10.45 (0.1) | |
| 0.5 | DDIM-GMM-ORTHO-VUB$^*$ | **28.73 (10)** | 17.89 (1) | 11.92 (1) | 10.81 (1) | |
| 1.0 | DDIM | 81.88 | 37.09 | 15.45 | 11.33 | |
| 1.0 | DDIM-GMM-RAND | **79.93** | **35.85 (10)** | 15.17 (10) | 11.50 (1) | |
| 1.0 | DDIM-GMM-ORTHO | **80.18 (10)** | **35.93 (10)** | 15.03 (10) | 11.51 (1) | |
| 1.0 | DDIM-GMM-ORTHO-VUB | **69.25 (10)** | **23.88 (1)** | 15.44 (0.1) | 11.44 (1) | |
| 1.0 | DDIM-GMM-ORTHO-VUB$^*$ | **67.72 (10)** | **23.76 (10)** | 15.56 (1) | 11.38 (0.1) | |
| 1.0 | DDPM | | | | | 9.69 |

Table 9: FFHQ (FID↓)

### A.14.3 ImageNet

The FID and IS results with classifier and classifier-free guidance are in Tables 10-11 and Tables 12-13 respectively.

| Steps | | 10 | | 100 | | 1000 | |
|---|---|---|---|---|---|---|---|
| Guidance Scale | | 1 | 10 | 1 | 10 | 1 | 10 |
| $\eta$ | | | | | | | |
| 0 | DDIM | 21.94 | 15.65 | 10.78 | 11.66 | | |
| 0 | DDIM-GMM-RAND | 21.94 | **11.32** | 11.14 | 11.28 | | |
| 0 | DDIM-GMM-ORTHO | 22.00 | **11.26** | 11.22 | 11.28 | | |
| 0 | DDIM-GMM-ORTHO-VUB | 22.00 | **11.26** | 11.24 | 11.28 | | |
| 0 | DDIM-GMM-ORTHO-VUB* | 21.95 | **11.54** | 10.81 | 11.31 | | |
| 0.2 | DDIM | 22.39 | 15.83 | 10.59 | 11.55 | | |
| 0.2 | DDIM-GMM-RAND | 22.25 | **13.10** | 10.50 | 11.58 | | |
| 0.2 | DDIM-GMM-ORTHO | 22.11 | **12.95** | 10.50 | 11.57 | | |
| 0.2 | DDIM-GMM-ORTHO-VUB | 22.43 | **11.29** | 10.79 | 11.20 | | |
| 0.2 | DDIM-GMM-ORTHO-VUB* | 22.39 | **11.50** | 10.50 | 11.24 | | |
| 0.5 | DDIM | 25.31 | 16.72 | 10.04 | 11.39 | | |
| 0.5 | DDIM-GMM-RAND | 25.39 | **15.36** | 9.93 | 11.25 | | |
| 0.5 | DDIM-GMM-ORTHO | 25.35 | **15.27** | 9.88 | 11.25 | | |
| 0.5 | DDIM-GMM-ORTHO-VUB | 24.72 | **11.79** | 9.94 | 11.01 | | |
| 0.5 | DDIM-GMM-ORTHO-VUB* | 24.96 | **11.93** | 9.91 | 10.96 | | |
| 1.0 | DDIM | 47.61 | 26.09 | 8.97 | 11.48 | | |
| 1.0 | DDIM-GMM-RAND | 46.55 | **23.71** | 8.94 | **10.31** | | |
| 1.0 | DDIM-GMM-ORTHO | 46.45 | **23.57** | 8.91 | **10.28** | | |
| 1.0 | DDIM-GMM-ORTHO-VUB | **37.97** | **18.60** | 8.82 | 11.38 | | |
| 1.0 | DDIM-GMM-ORTHO-VUB* | **38.08** | **18.68** | 8.80 | 11.40 | | |
| 1.0 | DDPM | | | | | 8.50 | 10.75 |

Table 10: Class-conditional ImageNet with classifier guidance(FID↓)

| Steps | | 10 | | 100 | | 1000 | |
|---|---|---|---|---|---|---|---|
| Guidance Scale | | 1 | 10 | 1 | 10 | 1 | 10 |
| $\eta$ | | | | | | | |
| 0 | DDIM | 66.15 | 136.76 | 98.71 | 179.62 | | |
| 0 | DDIM-GMM-RAND | 66.26 | **161.19** | 98.34 | **181.43** | | |
| 0 | DDIM-GMM-ORTHO | 66.16 | **161.47** | 97.25 | **181.33** | | |
| 0 | DDIM-GMM-ORTHO-VUB | 66.17 | **161.45** | 97.14 | **181.33** | | |
| 0 | DDIM-GMM-ORTHO-VUB* | 66.48 | **154.64** | 98.75 | 180.60 | | |
| 0.2 | DDIM | 66.17 | 135.01 | 99.4 | 179.86 | | |
| 0.2 | DDIM-GMM-RAND | 69.51 | **152.48** | 99.60 | **182.41** | | |
| 0.2 | DDIM-GMM-ORTHO | **69.48** | **152.10** | 100.12 | **182.25** | | |
| 0.2 | DDIM-GMM-ORTHO-VUB | 65.35 | **162.13** | 99.00 | **183.91** | | |
| 0.2 | DDIM-GMM-ORTHO-VUB* | 66.50 | **156.31** | 99.64 | **182.60** | | |
| 0.5 | DDIM | 62.42 | 131.49 | 105.02 | 190.45 | | |
| 0.5 | DDIM-GMM-RAND | 62.25 | **138.43** | 106.46 | 191.35 | | |
| 0.5 | DDIM-GMM-ORTHO | 62.53 | **139.64** | **106.18** | 190.87 | | |
| 0.5 | DDIM-GMM-ORTHO-VUB | **67.45** | **161.57** | **106.74** | **192.72** | | |
| 0.5 | DDIM-GMM-ORTHO-VUB* | **65.35** | **156.91** | **106.46** | **192.17** | | |
| 1.0 | DDIM | 35.19 | 86.76 | 116.59 | 207.78 | | |
| 1.0 | DDIM-GMM-RAND | 36.78 | **95.47** | 116.54 | 207.52 | | |
| 1.0 | DDIM-GMM-ORTHO | **37.02** | **96.90** | 117.09 | 208.34 | | |
| 1.0 | DDIM-GMM-ORTHO-VUB | **48.74** | **120.63** | 117.19 | **209.36** | | |
| 1.0 | DDIM-GMM-ORTHO-VUB* | 35.03 | **118.57** | 117.52 | 208.74 | | |
| 1.0 | DDPM | | | | | 118.28 | 210.53 |

Table 11: Class-conditional ImageNet with classifier guidance(IS↑)

| Steps | | 10 | | 100 | | 1000 | |
|---|---|---|---|---|---|---|---|
| Guidance Scale | | 2.5 | 5 | 2.5 | 5 | 2.5 | 5 |
| $\eta$ | | | | | | | |
| 0 | DDIM | 10.15 | 18.56 | 9.57 | 19.94 | | |
| 0 | DDIM-GMM-RAND | **6.90** | **16.77** | 9.23 | 19.83 | | |
| 0 | DDIM-GMM-ORTHO | **6.90** | **16.64** | 9.20 | 19.84 | | |
| 0 | DDIM-GMM-ORTHO-VUB | **6.94** | **16.61** | 9.13 | 19.66 | | |
| 0 | DDIM-GMM-ORTHO-VUB* | **6.72** | **16.14** | 9.22 | 19.77 | | |
| 0.2 | DDIM | 10.35 | 18.60 | 9.67 | 20.29 | | |
| 0.2 | DDIM-GMM-RAND | **8.52** | 17.90 | 9.70 | 20.17 | | |
| 0.2 | DDIM-GMM-ORTHO | **8.54** | 17.85 | 9.73 | 20.18 | | |
| 0.2 | DDIM-GMM-ORTHO-VUB | **7.06** | **16.78** | 9.27 | 20.04 | | |
| 0.2 | DDIM-GMM-ORTHO-VUB* | **6.75** | **16.49** | 9.38 | 20.05 | | |
| 0.5 | DDIM | 11.15 | 19.13 | 10.70 | 21.41 | | |
| 0.5 | DDIM-GMM-RAND | 10.28 | 18.61 | 10.60 | **15.73** | | |
| 0.5 | DDIM-GMM-ORTHO | 10.29 | 18.54 | 10.56 | **15.73** | | |
| 0.5 | DDIM-GMM-ORTHO-VUB | **7.59** | **17.91** | 10.27 | 21.10 | | |
| 0.5 | DDIM-GMM-ORTHO-VUB* | **7.44** | **17.51** | 10.38 | 21.17 | | |
| 1.0 | DDIM | 17.50 | 20.42 | 12.97 | 23.56 | | |
| 1.0 | DDIM-GMM-RAND | **15.95** | 19.91 | **10.84** | **22.38** | | |
| 1.0 | DDIM-GMM-ORTHO | **15.94** | 19.92 | **10.86** | **22.26** | | |
| 1.0 | DDIM-GMM-ORTHO-VUB | **12.38** | 19.70 | 12.80 | 22.94 | | |
| 1.0 | DDIM-GMM-ORTHO-VUB* | **12.34** | 19.64 | 12.80 | 23.59 | | |
| 1.0 | DDPM | | | | | 12.61 | 23.38 |

Table 12: Class-conditional ImageNet with classifier free guidance(FID↓)

| Steps | | 10 | | 100 | | 1000 | |
|---|---|---|---|---|---|---|---|
| Guidance Scale | | 2.5 | 5 | 2.5 | 5 | 2.5 | 5 |
| $\eta$ | | | | | | | |
| 0 | DDIM | 196.73 | 296.89 | 237.98 | 321.59 | | |
| 0 | DDIM-GMM-RAND | **209.90** | **318.73** | 238.63 | 321.68 | | |
| 0 | DDIM-GMM-ORTHO | **207.65** | **320.66** | 238.97 | **322.90** | | |
| 0 | DDIM-GMM-ORTHO-VUB | **207.85** | **319.13** | **240.88** | **323.84** | | |
| 0 | DDIM-GMM-ORTHO-VUB* | **200.89** | **316.90** | 238.97 | 321.40 | | |
| 0.2 | DDIM | 196.15 | 298.19 | 241.48 | 323.87 | | |
| 0.2 | DDIM-GMM-RAND | **209.22** | **311.00** | 239.52 | 323.23 | | |
| 0.2 | DDIM-GMM-ORTHO | **209.02** | **313.10** | 238.85 | 322.96 | | |
| 0.2 | DDIM-GMM-ORTHO-VUB | **208.91** | **318.41** | **243.14** | **326.02** | | |
| 0.2 | DDIM-GMM-ORTHO-VUB* | **204.41** | **318.19** | 241.09 | **324.86** | | |
| 0.5 | DDIM | 193.44 | 297.65 | 250.97 | 330.86 | | |
| 0.5 | DDIM-GMM-RAND | **198.14** | **306.79** | 251.02 | 331.14 | | |
| 0.5 | DDIM-GMM-ORTHO | **197.84** | **305.75** | 250.27 | 331.16 | | |
| 0.5 | DDIM-GMM-ORTHO-VUB | **211.93** | **319.87** | **251.72** | **333.34** | | |
| 0.5 | DDIM-GMM-ORTHO-VUB* | **210.79** | **320.60** | **251.74** | **333.29** | | |
| 1.0 | DDIM | 148.96 | 281.86 | 272.39 | 345.19 | | |
| 1.0 | DDIM-GMM-RAND | **157.54** | **294.47** | 270.73 | **354.69** | | |
| 1.0 | DDIM-GMM-ORTHO | **158.41** | **294.03** | 270.28 | **355.00** | | |
| 1.0 | DDIM-GMM-ORTHO-VUB | **187.49** | **309.10** | **273.73** | **349.01** | | |
| 1.0 | DDIM-GMM-ORTHO-VUB* | **185.85** | **310.05** | 271.89 | **346.74** | | |
| 1.0 | DDPM | | | | | 277.40 | 349.48 |

Table 13: Class-conditional ImageNet with classifier free guidance(IS↑)