# OpenReview forum: "Improved DDIM Sampling with Moment Matching Gaussian Mixtures"
_TMLR — Accepted by TMLR_

### Review · Reviewer_k1fx · 2025-09-16

**Summary Of Contributions:**

The paper improves DDIM sampling by introducing a moment-matched Gaussian mixture kernel in place of the standard Gaussian reverse kernel, enabling multimodal modeling in the denoising step. The method includes several parameterization schemes, extends to rectified flow models, and shows improved FID/IS, particularly in low-step sampling regimes.

The approach provides a principled extension of DDIM, demonstrates broad applicability across tasks, and achieves strong empirical gains with few sampling steps.

The improvements diminish with many steps, and the method may add implementation complexity.

**Audience:**

Yes

**Audience Explanation:**

The findings would be of interest to researchers working on diffusion models, generative modeling efficiency, and sampling algorithms, as the proposed method improves sample quality in low-step regimes and extends to rectified flow models.

**Broader Impact Concerns:**

The broader impact statement in the manuscript is sufficient.

**Claims And Evidence:**

Yes

**Claims Explanation:**

The claims are generally supported by empirical evidence across a range of datasets and tasks, with improvements in FID and IS clearly demonstrated, particularly in low-step regimes. The experiments are comprehensive, covering unconditional, class-conditional, and text-to-image generation, and the comparisons to DDIM baselines are convincing. However, the theoretical justification for why the Gaussian mixture approach provides improvements is less developed, and some claims about broader applicability would benefit from deeper analysis or ablation studies.

**Requested Changes:**

Clarify the uniqueness and motivation of the Gaussian mixture model approach. Provide a clearer theoretical explanation or intuition for why moment-matched mixtures improve over Gaussian kernels, beyond empirical evidence.

Add ablation studies to show the effect of different parameterization schemes (RAND, ORTHO, ORTHO-VUB) on performance and stability.

Typo: $\delta_k^t$ should be $\delta_t^k$ in Section 3.1

---

> ### Author Response · Authors · 2025-10-01
> **Response to Reviewer k1fx**
>
> Thank you for the review and valuable feedback. Please see our responses below to your specific requests.
>
> **Motivation and Uniqueness**
>
> The proposed approach of using a multimodal (GMM) kernel within the DDIM framework is motivated by the need to model multimodality in the true denoising distributions $q(x_{t-1}|x_t)$ of diffusion processes, especially when the step sizes $t\rightarrow(t-1)$ are large [1, 2]. We have tried to provide a justification in Eq. 14 (Sec. 3.2) on how using a multimodal DDIM-GMM kernel $q_{\sigma, \mathcal{M}}(x_{t-1}|x_t, x_0)$ can allow more flexibility in modeling the true denoising distributions relative to a unimodal DDIM kernel $q_{\sigma}(x_{t-1}|x_t, x_0)$ [3]. We have also tried to make this motivation clear in the Introduction (beginning of page 2). As discussed in the same paragraph, satisfying marginal constraints when sampling with a multimodal kernel is not straightforward. We derive constraints on the GMM parameters that allow us to match the first and second order central moments of the marginals instead. With these constraints, the marginals at every time point $t$ along the sampling trajectory have the same moments as the true marginals of the diffusion process. The proof of this is provided in Appendix A.1. Our empirical results show that this is sufficient to generate samples of better quality relative to DDIM when using fewer sampling steps, with a proper choice of GMM parameters (#components $K$, offsets $\delta$'s and their scale $s$). We believe that sampling with moment matching GMM kernels and deriving the parameter constraints is unique to our proposed approach.
>
> **Theoretical justification**
>
> An intuitive explanation for why Gaussian mixtures can provide potential benefits over a unimodal Gaussian kernel is provided in Sec. 3.2 and Sec. 4.1 for Diffusion and Rectified Flow models respectively. We start by assuming an analytical form for real-world dataset densities as mixtures of Diracs or Gaussians centered around data points. Using Bayesian calculus, we show that plugging in a multimodal kernel yields a more expressive parametric denoising distribution. In the simplest case, the DDIM-GMM kernel with a single mixture component ($K=1$) and the proposed constraints reduces to the DDIM kernel. We believe this provides an intuition from a variational function modeling perspective.
>
> In Appendix A.4, we derive an upper bound on the ELBO loss function for training a (hypothetical) denoiser if the forward diffusion process were designed to follow the marginals induced by the proposed DDIM-GMM kernel. It turns out that this upper bound has a form that resembles an augmented version of the Gaussian diffusion (DDPM) loss function ($L_{simple,w}$) with a different weighting $w$. Specifically, the augmentation uses noisy samples from the DDIM-GMM forward marginals instead of the DDIM Gaussian marginals (=DDPM marginals). We believe this provides another intuitive justification of why sampling from an appropriately parameterized DDIM-GMM kernel makes sense. The same upper bound can be potentially used to further fine-tune a pre-trained diffusion model to work better with the DDIM-GMM sampler but we leave this exploration for future work.
>
> **Performance and Stability Ablations with RAND, ORTHO and ORTHO-VUB**
>
> The two main parameters affecting the stability and performance of the different DDIM-GMM variants are the offset scale ($s$) and the number of Gaussian mixture components ($K$). We included ablation studies with these hyperparameters for the ORTHO-VUB variant in Appendix A.8 in the original submission. We have conducted similar ablations for the other variants and will incorporate these results in a revised version of the manuscript.
>
> Thank you for pointing out the typo. We will incorporate the correction in the next version.
>
> **References**
> 1. Hanzhong Guo et. al., Gaussian Mixture Solvers for Diffusion Models, NeurIPS 2023.
> 2. Zhisheng Xiao et. al., Tackling the Generative Learning Trilemma with Denoising Diffusion GANs, ICLR 2022.
> 3. Song et. al., Denoising diffusion implicit models, ICLR 2021.

---

### Review · Reviewer_annb · 2025-11-16

**Summary Of Contributions:**

The authors propose a new sampling strategy for pre-trained diffusion models where the reverse kernel $p(x_{t-1} | x_{t}, x_{0})$ is a Gaussian mixture rather than a Gaussian. The parameters of this Gaussian mixture model are randomly chosen, at each sampling step, such that they have the same mean and covariance as the conventional Gaussian reverse kernel used in diffusion models. The authors provide three different ways to choose the parameters randomly subject to this moment matching constraint with the DDIM kernel, and several more for flow-matching/rectified flow kernels (generated by the interpolation. Finally, the authors conduct experiments on real data to compare these new methods to baseline diffusion models; their experiments are conducted on models trained from scratch using the standard denoising/flow-matching losses.

**Audience:**

Yes

**Audience Explanation:**

Diffusion models are the premier generative modeling paradigm for several modalities. Improving sampling efficiency in pre-trained diffusion models is a currently popular area of research. This paper fits neatly within the aforementioned research goal, so I am confident that this work in this area would be interesting/relevant to some of TMLR's audience.

**Claims And Evidence:**

Yes

**Claims Explanation:**

My understanding is that the main claims made in the submission can be summarized as follows:

> C1: We can parameterize the reverse kernel in a diffusion sampler to be a GMM, not a Gaussian, and can match the first two moments of this GMM with the conventional Gaussian kernel.

This is a mathematical fact, and in my opinion is adequately justified by the constructions in Section 3.1 and related proofs in A.1, A.2, A.3.

> C2: Using this reverse kernel for diffusion sampling enables better sampling at small step counts.

This is proven, to a good extent, by some of the experiments (e.g. Figures 1-4), which generally show either convergence to the usual DDPM sampler's performance as the step count increases, and improvement at fixed small step count. There are several datasets and sampling settings used (unconditional CelebAHQ, unconditional FFHQ, class-conditional ImageNet, text-to-image COYO-700M).

The difference between methods is much more visible when using pre-trained text-to-image models than models trained by the authors. This is a great sign for the method, but it poses a confounding variable: is this difference brought on by different training, bigger scale, or just that the methodology just works much better for text-to-image? This confounding variable is not addressed, which poses a slight caveat to claim C2.

On the whole, however, C2 seems well-supported.

**Requested Changes:**

Overall I think the paper seems good, and so none of the below changes are essential to secure my recommendation for acceptance, but addressing them would improve the paper.

- In relation to the caveat about Claim C2, can you either train a text-to-image model or use a pre-trained model (say the original DiT, which has pre-trained weights available online, to do class-conditional ImageNet sampling, or one of many of pre-trained unconditional models) to add experiments similar to those conducted in Figures 1 and 2?
  - Similarly w.r.t. the experiments, for the CFG guidance parameter, can you explain how it was chosen? It seems to vary across the experiments. Was there any kind of cherry picking (as this would partially explain the disparity between unconditional and text-conditional models' behavior discussed above)?

- In addition to the number of function evaluations, the FLOPs consumed or the latency consumed by the proposed methods vs DDPM/DDIM/RF baselines would be interesting to track, and improvements here would make this work more meaningful to the community.

- The math notation makes it relatively hard to parse several of the terms (e.g. A.3 has the GMM() function which ideally should be written out). Similarly it alternates between treating $t$ as an integer and as a real number. Personally it would be really great if there were notation to tell at a glance which setting each calculation is restricted to (e.g., general calculation, only applying for DDPM kernel, or only applying for RF/FM kernel). An alternative is that you try to package your results into theorems where you can precisely define everything and keep it self-contained; this edit would also help the organization of your paper a great deal, in my opinion.

---

> ### Author Response · Authors · 2025-12-13
> **Response to Reviewer annb (Part 1)**
>
> We sincerely thank you for the feedback. Please see below for answers to your specific questions.
>
> **Results with Pre-trained Models (Part 1)**
>
> Thank you for the insightful observation on the differences between relative gains observed in pretrained text-to-image models versus the models that we trained ourselves. Following the suggestion, we conducted additional experiments with pre-trained models. Specifically, we used the unconditional model trained on CelebAHQ [1, 2] and the class-conditional DiT-XL/2 model trained on ImageNet [3, 4], both at 256x256 resolution. The results from our experiments are listed below.
>
> >CelebAHQ Pre-trained Model [2]: FID
>
>   | **η=0.0** | **10** | **20** | **50** | **100** | | **η=0.2** | **10** | **20** | **50** | **100** |
>   |---|---|---|---|---|---|---|---|---|---|---|
>   | DDIM | 23.63 | 13.87 | 10.81 | 10.30 | | DDIM | 24.36 | 14.03 | 10.77 | 10.35 |
>   | DDIM-GMM-RAND | 17.99 | 13.77 | 10.57 | 10.13 | | DDIM-GMM-RAND | 21.22 | 11.86 | 10.59 | 10.08 |
>   | DDIM-GMM-ORTHO | 18.19 | 13.78 | 10.51 | 10.13 | | DDIM-GMM-ORTHO | 21.34 | 12.01 | 10.58 | 10.08 |
>   | DDIM-GMM-ORTHO-VUB | 18.19 | 13.78 | 10.51 | 10.13 | | DDIM-GMM-ORTHO-VUB | 17.88 | 14.30 | 10.45 | 10.08 |
>
>   | **η=0.5** | **10** | **20** | **50** | **100** | | **η=1.0** | **10** | **20** | **50** | **100** |
>   |---|---|---|---|---|---|---|---|---|---|---|
>   | DDIM | 28.60 | 16.40 | 11.56 | 10.71 | | DDIM | 59.21 | 32.83 | 17.75 | 13.65 |
>   | DDIM-GMM-RAND | 25.74 | 14.63 | 10.33 | 10.78 | | DDIM-GMM-RAND | 54.12 | 29.48 | 15.50 | 12.19 |
>   | DDIM-GMM-ORTHO | 25.55 | 14.59 | 10.29 | 10.77 | | DDIM-GMM-ORTHO | 54.39 | 29.87 | 15.55 | 12.01 |
>   | DDIM-GMM-ORTHO-VUB | 19.73 | 16.37 | 11.39 | 10.66 | | DDIM-GMM-ORTHO-VUB | 46.83 | 19.31 | 17.43 | 13.41 |
>
>
> The results above suggest that the FIDs using the pretrained model are better relative to the models we trained (Table 7).
> We also see that the DDIM-GMM variants consistently outperform DDIM, similar to the results reported in the paper.
>
> **Latency of DDIM-GMM relative to DDIM**
>
> We briefly discuss the extra computational overhead involved in using the proposed approach in Section 3.1.3, towards the end.
> Most of the overhead is in precomputing the GMM parameters prior to sampling and saving them in memory.
> Appendix A.10 discusses a few different strategies to save on compute and storage based on empirical evidence.
>
> In order to further quantify the overhead, we conducted a small experiment using the proposed samplers. This experiment was run on a Nvidia Tesla V100 GPU (32GB memory) with the DDIM-GMM-* samplers using 8 clusters. Using a model trained on the CelebAHQ dataset, we report the average time taken by each sampler, run for 10 sampling steps, per sample (averaged across 100 samples).
>
> DDIM: 261ms \
> DDIM-GMM-RAND: 281ms\
> DDIM-GMM-ORTHO: 276ms\
> DDIM-GMM-ORTHO-VUB: 268ms
>
> The DDIM-GMM-* samplers incur a small overhead during sampling relative to DDIM. This overhead is higher for DDIM-GMM-RAND and DDIM-GMM-ORTHO due to explicitly computing the covariance matrix (Eq. 8). DDIM-GMM-ORTHO-VUB is relatively faster due to not requiring the covariance matrix to be computed explicitly.
>
> In addition to the sampling overhead, there is a small overhead during initialization phase, which are reported below.
>
> DDIM-GMM-RAND: 0.9ms\
> DDIM-GMM-ORTHO-*: 590ms
>
> The larger initialization time for DDIM-GMM-ORTHO-* samplers is due to the SVD operation.
>
> **Notation**
>
> Thank you for the suggestions on improving the mathematical notations in the paper.
>
> We will explicitly write out the form of the GMMs in the forward process equation of A.3.
> We will also change the notation of the time variable to distinguish between continuous and discrete contexts.
> Specifically, we will change the variable notation to *τ* to refer to continuous time in the Rectified Flow Matching sections.
> All of the above changes will be incorporated in the updated version.

---

> ### Author Response · Authors · 2025-12-13
> **Response to Reviewer annb (Part 2)**
>
> **Results with Pre-trained Models (Part 2)**
>
> >ImageNet Pre-trained Model (DiT-XL/2) [4], No CFG: FID/IS
>
>   | **η=0.0** | **10** | **100** | | **η=0.2** | **10** | **100** |
>   |---|---|---|---|---|---|---|
>   | DDIM | 32.83/67.65 | 8.43/115.80 | | DDIM | 33.60/66.43 | 8.33/116.35 |
>   | DDIM-GMM-ORTHO-VUB | 32.37/68.02 | 8.40/116.26 | | DDIM-GMM-ORTHO-VUB | 33.42/67.26 | 7.28/122.33 |
>
>   | **η=0.5** | **10** | **100** | | **η=1.0** | **10** | **100** |
>   |---|---|---|---|---|---|---|
>   | DDIM | 39.67/59.58 | 7.85/120.47 | | DDIM | 76.12/26.02 | 7.15/127.24 |
>   | DDIM-GMM-ORTHO-VUB | 39.38/60.22 | 7.07/124.18 | | DDIM-GMM-ORTHO-VUB | 76.24/25.75 | 7.06/128.92 |
>
> The DiT work [3] reports results (Table 2) with no guidance for the DiT-XL/2 model. We chose to use the same setting and experimented with the DDIM-GMM-ORTHO-VUB sampler. Using a small scale search over the sampler hyperparameters, there exists a setting that performs better than DDIM, in almost all the cases above. However, the relative improvements are smaller compared to the unconditional models, text-to-image model or ImageNet models we trained and sampled with CFG guidance. We also used our model to sample without any guidance and observed similar results below.
>
> ImageNet Model (UNet), No CFG: FID/IS
>
>   | **η=0.0** | **10** | **100** | | **η=0.2** | **10** | **100** |
>   |---|---|---|---|---|---|---|
>   | DDIM | 33.32/39.87 | 18.04/59.34 | | DDIM | 33.92/39.57 | 17.72/59.69 |
>   | DDIM-GMM-ORTHO-VUB | 33.17/40.30 | 17.70/59.86 | | DDIM-GMM-ORTHO-VUB | 33.98/39.89 | 17.49/60.39 |
>
>   | **η=0.5** | **10** | **100** | | **η=1.0** | **10** | **100** |
>   |---|---|---|---|---|---|---|
>   | DDIM | 38.68/36.43 | 17.02/61.63 | | DDIM | 67.97/19.34 | 15.49/66.84 |
>   | DDIM-GMM-ORTHO-VUB | 38.42/37.76 | 17.03/62.26 | | DDIM-GMM-ORTHO-VUB | 59.50/25.93 | 15.34/67.22 |
>
> The above is possibly an effect of how DDIM and CFG interact [10], which is not fully understood to the best of our knowledge. A possible explanation could be based on recent works on Diffusion model generalization [5, 6], which suggest that model underfitting is favorable for generalization. Specifically, there seem to be distinct regions in  latent space $x_t$ [5], where the score matching loss leads to model underfitting (generalization) or overfitting (memorization). We conjecture that ImageNet models are trained long enough to closely approximate the true empirical conditional score in some $x_t$ subspace. For instance, the DiT-XL/2 model is trained for at least 400K steps with batch size 256 and our model is trained for more than 500K steps with batch size 128. However, the unconditional score is not equally well trained due to the lower sampling proportion (0.1) of its conditioning latent during training. The contribution from the underfit unconditional score model is switched off with no CFG, possibly increasing the likelihood of sampling from the memorized portions of latent space, narrowing the differences between DDIM and DDIM-GMM.
>
> We would also like to highlight the difference between the conditioning latents in ImageNet vs. text-to-image sampling. In the case of ImageNet models, the conditioning latents are fixed after training and reused during sampling. On the other hand, the conditioning latents are a function of the prompts in text-to-image models, which are not necessarily the same as the ones used during training. Based on the generalization/memorization and latent space exploration hypothesis, this could be a potential source of difference between the relative gains of the DDIM-GMM samplers over DDIM in the two cases.
>
> **CFG Scale**
>
> The CFG scale of 7.5 was chosen for text-to-image generation with the Stable Diffusion v2.1 model in line with community best practices [7, 8].
>
> For ImageNet experiment, we chose CFG scales (2.5 and 5.0) to include both a low and a high value based on the results in the original CFG work [9].
> We hope that the new experiments help further inform the expected improvements under different CFG regimes for ImageNet models.
>
> **References**
> 1. R. Rombach et. al., High-Resolution Image Synthesis with Latent Diffusion Models, CVPR 2022.
> 2. https://github.com/CompVis/latent-diffusion?tab=readme-ov-file#model-zoo
> 3. W. Peebles and S. Xie, Scalable Diffusion Models with Transformers, ICCV 2023.
> 4. https://github.com/facebookresearch/DiT?tab=readme-ov-file
> 5. T. Bonnaire et. al., Why Diffusion Models Don’t Memorize: The Role of Implicit Dynamical Regularization in Training, NeurIPS 2025.
> 6. Selective Underfitting in Diffusion Models, arXiv:2510.01378v1, 2025.
> 7. https://github.com/CompVis/stable-diffusion
> 8. https://huggingface.co/blog/stable_diffusion
> 9. J. Ho and T. Salimans, Classifier-free Diffusion Guidance, arXiv:2207.12598v1, 2022.
> 10. A. Bradley and P. Nakkiran, Classifier-Free Guidance is a Predictor-Corrector, arXiv:2408.09000v2, 2024.

---

### Review · Reviewer_e4u2 · 2026-02-19

**Summary Of Contributions:**

The authors propose a new method to accelerate sampling of diffusion models. The key idea is to parametrize the conditional distribution of the reverse process as a Gaussian Mixture instead of a pure Gaussian. The authors select the mixture model such that it matches the first two moments of the usual Gaussian conditional. The authors show improved results in the low NFEs regime over the DDIM sampler across different datasets and models.

**Audience:**

Yes

**Audience Explanation:**

The paper is on accelerating sampling of diffusion models, which is an area of significant interest. That said:

- Despite the fact that the paper shows improvement over DDIM, for low NFEs, the produced samples are actually not of great quality. Even with 20 steps, we see an FID in the order of 20 for simple datasets such as CelebA. In other words, the regime at which this work is actually significantly better than DDIM is a regime where results are not great either way, hinting towards limited applicability of this problem in the real world.

**Claims And Evidence:**

No

**Claims Explanation:**

- The authors model the reverse conditional with a Gaussian Mixture. However, as the step size becomes smaller and smaller, the true reverse conditional approximates a pure Gaussian. While it is true that this does not hold for larger step sizes, the choice of a Gaussian Mixture is heuristic. There is no clear evidence on why the reverse conditional for larger step sizes should have this particular form.
- Even if we accept the Gaussian Mixture form as a reasonable approximation for the reverse conditional when the step size is large, it would be more reasonable for the centers of these Gaussians to differ in a more "interesting" way than the one explored by the authors. In the GMM parameters section (3.1), the authors explore random variations around the same mean. While I understand that this is done to avoid retraining of a network, I do believe that this approximation is far from how the reverse conditional truly looks like as the step size increases.
- The experimental evaluation of the paper is pretty limited in terms of baselines. The authors compare only with DDIM. There are several other acceleration methods (as the authors acknowledge in their Related Work section) that achieve much better results than DDIM. Avoiding comparison with such methods makes it hard to evaluate the significance of this work.

**Requested Changes:**

- The authors should include comparisons with other methods for accelerating diffusion models, listed in the Related Work section.

---

> ### Author Response · Authors · 2026-03-05
> **Response to Reviewer e4u2 (Part 1)**
>
> We thank the reviewer for the feedback. Please see our responses below to your specific concerns.
>
> **GMM form as a principled choice**
>
> We agree with the reviewer that the true reverse conditional approaches a Gaussian as the step size decreases. However, the choice of a Gaussian Mixture is not purely heuristic for larger step sizes. We offer two
> complementary justifications:
>
> First, recent works [1, 2] have shown empirically and theoretically that the true reverse conditional $q(x_{t-1}|x_t)$ is multimodal when the step size is large. This directly motivates a multimodal parametric form. We show
> in Sec. 3.2 that the proposed DDIM-GMM kernel yields a denoising distribution $q_{\sigma, \mathcal{M}}(x_{t-1}|x_t)$ that is a mixture of Gaussians — strictly more expressive than the unimodal distribution produced by the
> DDIM kernel. This holds even when the underlying data distribution is a mixture of Gaussians, not just a sum of Diracs.
>
> Second, in Appendix A.5 we derive an upper bound on the ELBO loss under the DDIM-GMM inference process. This upper bound takes the form of an augmented version of the standard DDPM $\mathcal{L}_{simple,w}$ loss, with modified
> weights and noisy samples drawn from the DDIM-GMM forward marginals. This provides a principled training-free justification: a pre-trained DDPM denoiser already approximately minimizes a loss whose augmented form corresponds
> to sampling with the DDIM-GMM kernel.
>
> **On the choice of GMM centers**
>
> The reviewer notes that random variations around the same mean may be far from the true reverse conditional. We agree this is a simplification, but highlight two points. First, the key insight is not that the centers are
> "interesting" — it is that the moment-matching constraint ensures the marginal distribution at every step remains consistent with the DDPM forward process. Any choice of offsets satisfying this constraint is valid, and
> random/orthogonal directions provide a principled, training-free way to satisfy it. Second, our empirical results show that even this simple parametrization yields consistent improvements, which suggests that the
> moment-matching constraint itself — rather than the specific geometry of the centers — is the key driver of the gains. We note that the proposed moment-matching framework naturally opens up possibilities for designing
> constrained optimization approaches to find "more interesting" sets of offsets — for instance by maximizing a sample quality metric such as FID or KID on a training set — that could potentially boost performance further. We
> leave this exploration for future work.

---

> ### Author Response · Authors · 2026-03-05
> **Response to Reviewer e4u2 (Part 2)**
>
> **Comparison with other acceleration methods**
>
> We agree that comparison with methods beyond DDIM strengthens the evaluation. We have added a comparison with DPM-Solver [3] in Appendix A.11. On the class-conditional ImageNet model with classifier-free guidance,
> DDIM-GMM-ORTHO-VUB outperforms DPM-Solver on the FID metric in all settings (10 and 100 steps, guidance scales 2.5 and 5), as shown in Table 5 in Appendix A.11.
>
> We note that a direct comparison with distillation-based methods (e.g., consistency models, progressive distillation) is outside the intended scope of this work: those methods require additional training on top of a
> pre-trained model, whereas the proposed approach is entirely training-free. The focus of this paper is specifically on improving the DDIM sampler — one of the most widely used training-free samplers — through a principled
> moment-matching extension. Our work is in fact complementary to distillation approaches, as an improved DDIM sampler can serve as a better teacher [4, 5]. Similarly, higher-order ODE solvers such as DPM-Solver++ [6] address
> an orthogonal axis of improvement (solver order vs. kernel expressiveness) and a combined approach could be of interest, but is beyond the scope of this paper. We will be releasing our implementation alongside the paper,
> which will allow the community to directly compare our approach with other methods and build upon it.
>
> **Limited applicability concern**
>
> The reviewer notes that FIDs of ~20 at 20 steps on CelebA are not practically useful. We respectfully disagree with this framing for two reasons.
>
> First, this does not account for the text-to-image results. Using Stable Diffusion v2.1 on COYO-700M (30k pairs), DDIM-GMM consistently and significantly improves FID and IS over DDIM with 5 and 10 sampling steps — a
> practically relevant setting for large-scale generation where FID values are useful in realistic applications, see for e.g. images in Fig. 12. Furthermore, absolute FID values depend strongly on the quality of the underlying
> denoising model: as evidenced by the newer experimental results with pre-trained models reported in response to other reviewers below, stronger models yield substantially better absolute FIDs while the relative improvements
> from DDIM-GMM are maintained.
>
> Second, the FID gap between DDIM and DDIM-GMM is precisely largest in the low-step regime where quality matters most for practical speed-quality tradeoffs. The convergence of FID at larger step counts is expected and reflects
> that DDIM itself becomes adequate as step count increases — it is not a limitation of the proposed approach.
>
> **References**
> 1. H. Guo et al., Gaussian Mixture Solvers for Diffusion Models, NeurIPS 2023.
> 2. Z. Xiao et al., Tackling the Generative Learning Trilemma with Denoising Diffusion GANs, ICLR 2022.
> 3. C. Lu et al., DPM-Solver: A Fast ODE Solver for Diffusion Probabilistic Model Sampling in Around 10 Steps, NeurIPS 2022.
> 4. T. Salimans and J. Ho, Progressive Distillation for Fast Sampling of Diffusion Models, ICLR 2022.
> 5. C. Meng et al., On Distillation of Guided Diffusion Models, CVPR 2023.
> 6. C. Lu et al., DPM-Solver++: Fast Solver for Guided Sampling of Diffusion Probabilistic Models, arXiv 2022.

---

### Author Response · Authors · 2026-05-20
**Camera-Ready Version Uploaded**

The camera-ready version has now been uploaded. We thank the Action Editor and the anonymous reviewers for their thoughtful and constructive feedback, which helped improve the quality of the work.

---

### Decision · Action_Editor_yoUV · 2026-05-04

**Recommendation:** Accept as is

**Audience:**

Yes

**Audience Explanation:**

Diffusion samplers are a timely topic in the AI community.

**Claims And Evidence:**

Yes

**Claims Explanation:**

The authors investigate a new sampling method of diffusion models, switching the classical Gaussian backward sampling with a mixture of Gaussian sampling. They describe a collection of methods to choose the (random) parameters of the mixture of Gaussians. They compare their new sampler against a few baselines for image generation at scale.